# Learning and Planning Multi-Agent Tasks via an MoE-based World Model

**Zijie Zhao**[1,2], **Zhongyue Zhao**[2], **Kaixuan Xu**[2], **Yuqian Fu**[1,2],
**Jiajun Chai**[3], **Yuanheng Zhu**[2,1,†], **Dongbin Zhao**[2,1,†]
[1]School of Artificial Intelligence, University of Chinese Academy of Sciences
[2]Institute of Automation, Chinese Academy of Sciences, [3]Meituan
{zhaozijie2022, zhaozhongyue2024, xukaixuan2023, fuyuqian2022}@ia.ac.cn
chaijiajun@meituan.com, {yuanheng.zhu, dongbin.zhao}@ia.ac.cn
[†]*Corresponding authors* *

## Abstract

Multi-task multi-agent reinforcement learning (MT-MARL) aims to develop a single model capable of solving a diverse set of tasks. However, existing methods often fall short due to the substantial variation in optimal policies across tasks, making it challenging for a single policy model to generalize effectively. In contrast, we find that many tasks exhibit **bounded similarity** in their underlying dynamics—highly similar within certain groups (e.g., door-open/close) diverge significantly between unrelated tasks (e.g., door-open & object-catch). To leverage this property, we reconsider the role of modularity in multi-task learning, and propose **M3W**, a novel approach that applies mixture-of-experts (MoE) to world model instead of policy, enabling both learning and planning. For learning, it uses a SoftMoE-based dynamics model alongside a SparseMoE-based predictor to facilitate knowledge reuse across similar tasks while avoiding gradient conflicts across dissimilar tasks. For planning, it evaluates and optimizes actions using the predicted rollouts from the world model, without relying directly on a explicit policy model, thereby overcoming the limitations of policy-centric methods. As the first MoE-based multi-task world model, M3W demonstrates superior performance, sample efficiency, and multi-task adaptability, as validated on Bi-DexHands with 14 tasks and MA-Mujoco with 24 tasks. The code are available at https://github.com/zhaozijie2022/m3w-marl.

## 1 Introduction

The goal of multi-task MARL is to develop a single model capable of solving multiple tasks, improving both learning efficiency and generalization performance [3]. The core challenge lies in reusing knowledge across similar tasks while avoiding gradient conflicts between dissimilar tasks [15]. Existing approaches attempted to introduce modularity into policy learning to address this issue, employing techniques such as subtask reorganization [36], skill discovery [42, 22], and network decomposition [49].

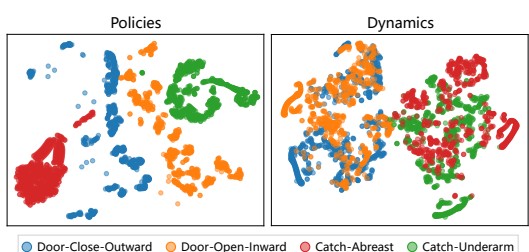

Figure 1: T-SNE visualization comparing the distributions of dynamics and policies.

*This work was supported in part by National Natural Science Foundation of China under Grants 62136008, 62293541, in part by Beijing Nova Program under Grant 20240484514, and in part by Beijing Natural Science Foundation under Grant No 4232056, L223020.

39th Conference on Neural Information Processing Systems (NeurIPS 2025).

However, we argue that existing policy-centric methods may fail to fully capture the nuanced relationship among tasks. As visualized in Figure 1, the policy distributions of the selected four tasks are mutually distinct. In contrast, we find that dynamics similarity exists within certain task groups (e.g., `door-open` and `door-close`; `catch-abreast` and `catch-underarm`), but not across different groups (e.g., between `door` and `catch` tasks). We refer to this phenomenon as **bounded similarity**, where tasks exhibit strong local similarity in dynamics within specific groups, but such similarity does not generalize globally across the entire task set.

This finding motivates two core insights. First, in terms of "*similarity*", modeling shared environment dynamics—rather than learning separate task-specific policies—can better exploit intra-group similarity and thus improve learning efficiency. Second, in terms of "*bounded*", a modular structure such as Mixture-of-Experts (MoE) [30] is needed to isolate dissimilar dynamics and prevent negative interference across tasks. Based on these insights, we reconsider the role of modularity in multi-task learning, and propose to apply it in the world model instead of the policy.

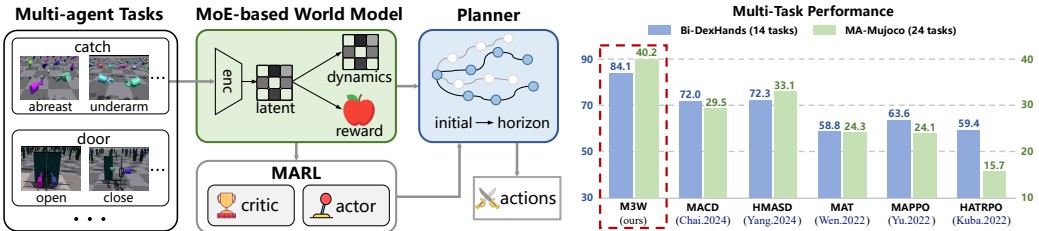

Figure 2: Overview. (*Left*) The framework of M3W. (*Right*) Performance comparisons.

A world model captures the feature of the environment, enabling the generation of imagined trajectories that can be used for model training to improve sample efficiency [10–12], and for planning to enhance performance [31, 14, 13]. As illustrated in Figure 2, we incorporate a MoE structure into the world model. Specifically, we construct a multi-agent dynamics model based on SoftMoE [28] and a reward predictor based on SparseMoE [30]. During execution, the framework bypasses an explicit policy model and directly evaluates and optimizes actions using predicted data generated by the world model. We name this approach as **M3W**, which employs a **M**oE-based **W**orld model for learning and planning in **M**ulti-agent **M**ulti-task scenarios. To the best of our knowledge, this is the first work to apply MoE in multi-task world models. The contributions of this paper are as follows:

1) We identify the phenomenon of bounded similarity in multi-task dynamics, and reconsider the role of modularity by applying it to world models rather than policies.

2) We propose a novel framework that learns dynamics and rewards via a MoE-based world model and performs planning directly on predicted rollouts, without relying on an explicit policy.

3) We evaluate M3W on Bi-DexHands and MA-Mujoco, demonstrating its superior performance, sample efficiency, and interpretability through extensive comparisons, ablation studies, and visualizations.

## 2  Related Work

**Multi-task MARL.** The key challenge in MT-MARL is how to reuse knowledge across similar tasks while avoiding gradient conflicts among dissimilar tasks [3]. Most existing methods address this issue from the perspective of policy learning [46, 18], exploring solutions such as subtask reorganization, skill discovery, and network decomposition. For example, UNMAS [2], TETQmix [49] design network architectures to accommodate the compositional variations of multi-agent system. ODIS [42], HMASD [40] learn coordination skills, enhancing generalization capabilities through skill combinations; DT2GS [36] employs a hierarchical structure to learn the decomposition and composition of subtasks in the multi-task set. As mentioned in Introduction, these policy-centric methods do not fully exploit the bounded similarity among tasks, and a model-based paradigm offers a more promising direction.

**Model-based MARL.** In multi-agent settings, world models often adopt communication-based paradigms to address unique challenges such as partial observability and locally non-stationary

dynamics [47]. For instance, AORPO[44] assumes agents can communicate to access opponents' actions, using this information to improve model learning. Building on this idea, MAMBA[7] incorporates Recurrent State-Space Models (RSSM) from DreamerV2[11] to construct a multi-agent world model. MACD[1] further introduces an explicit Transformer-based communication module, significantly enhancing model prediction accuracy. MA-TDMPC [14] and MAZero [21] extend TD-MPC [14, 13] and MuZero [31] to the multi-agent context.

**MoE in MTRL.** As model capacity increases and generalization becomes crucial, Mixture of Experts (MoE) has become a key approach in multi-task reinforcement learning (MTRL) for balancing shared and task-specific knowledge. Early works such as CARE [32] introduced context-based representations to differentiate tasks within a shared policy, while Paco [35] composed shared and task-specific parameters to enhance transfer. AMESAC [5] further incorporated attention-based expert selection to address task heterogeneity, and MOORE [16] imposed orthogonal constraints to encourage expert decoupling. Building on these, recent methods such as MCP [27], D2R [15], and M3 [24] employ MoE to enable hierarchical skill reuse, dynamic routing, and transferable policy learning. Unlike prior works that focus on MoE-based policy design, our approach integrates MoE into the world model, achieving modular learning of both dynamics and reward functions.

## 3   Problem Formulation

A cooperative MARL task can be modeled as a decentralized partially observable Markov decision process (dec-POMDP), defined by the tuple $\mathcal{M} = \langle \mathcal{I}, \mathcal{S}, \mathcal{O}, \mathcal{A}, \Omega, \mathcal{P}, \mathcal{R}, \gamma \rangle$, where $\mathcal{I} = \{1, \ldots, n\}$ is the set of agents, $\mathcal{S}$ is the global state space, $\mathcal{O} = \prod_{i=1}^{n} O^i$ is the joint observation space, $\mathcal{A} = \prod_{i=1}^{n} A^i$ is the joint action space, $\Omega : \mathcal{S} \times \mathcal{I} \to \mathcal{O}$ is the observation function, $\mathcal{P} : \mathcal{S} \times \mathcal{A} \to \mathcal{S}$ is the transition function, $\mathcal{R} : \mathcal{S} \times \mathcal{A} \to \mathbb{R}$ is the reward function, and $\gamma$ is the discount factor [34]. In single-task MARL, the goal is to learn a joint policy $\boldsymbol{\pi} = \prod_{i=1}^{n} \pi^i$ that maximizes the expected discounted return $\sum_{\tau=t}^{\infty} \gamma^\tau r_\tau$. In contrast, MT-MARL aims to train a single model that performs well across a finite set of tasks $\mathcal{T} = \{\mathcal{M}_k\}$, where each task $\mathcal{M}_k$ is itself a dec-POMDP with task-specific dynamics and reward functions.

**Multi-task Settings.** Following the general setting of multi-task MARL [43], we: (1) use one-hot encoding for each task as task label, (2) align dimensions of $\mathcal{S}_k$ and $\mathcal{O}_k$ across all tasks by padding zeros at end, (3) apply action space masks for $\mathcal{A}_k$; and (4) normalize or bound reward scales across tasks to facilitate stable learning.

## 4   Methodologies

As shown in Figure 2, M3W consists of two parts: a MoE-based world model and a multi-agent planner. During training, the world model is trained using real environment data to improve the quality of predicted rollouts. During execution, the planner evaluates and optimizes actions based on predictions from the world model.

### 4.1   Learning a World Model for MT-MARL

**Learning Multi-Agent Dynamics as Sequence Prediction.** At each timestep, the observation-action pairs $(o_t^i, a_t^i)$ from multiple agents can be arranged into a sequence [38], allowing the multi-agent dynamics to be modeled as a sequential prediction problem [1, 45]. The dynamics takes as input a sequence of length $n$, $[(o_t^1, a_t^1), (o_t^2, a_t^2), \ldots, (o_t^n, a_t^n)]$, and outputs a sequence of the same length, $[o_{t+1}^1, o_{t+1}^2, \ldots, o_{t+1}^n]$. This formulation treats multi-agent dynamics prediction as a structured sequence modeling problem.

**Multi-agent World Model.** Establishing dynamics and reward prediction within a shared latent space has been shown to effectively enhance the predictive capacity of world models [14, 10]. Following this principle, our world model operates entirely in the latent space and does not include a decoder. It comprises an encoder, a dynamics model, a reward predictor, and an actor-critic module. Consistent with existing model-based MARL approaches, we assume that agents can access the observations and actions of others, which can be achieved through a single round of communication in practice. To support multi-task generalization, we introduce a learnable task embedding $e$ for each task, which is used as a conditioning signal throughout the model. Let $z_t^i$ denote the latent state of agent $i$ at time

$t$, then the world model are defined as:

$$
\begin{aligned}
\text{Encoder:} \quad z_t^i &= p_E\left(o_t^i, e\right) \\
\text{Dynamics:} \quad \hat{z}_{t+1} &= q_D\left(\boldsymbol{z_t}, \boldsymbol{a_t}, e\right) \\
\text{Predictor:} \quad \hat{r}_t &= q_R\left(\boldsymbol{z_t}, \boldsymbol{a_t}, e\right) \\
\text{Critic:} \quad \hat{q}_t &= Q\left(\boldsymbol{z_t}, \boldsymbol{a_t}, e\right) \\
\text{Actor:} \quad \hat{a}_t^i &= \pi^{\text{Actor}}\left(\cdot \,\middle|\, z_t^i, e\right),
\end{aligned}
\tag{1}
$$

where $\boldsymbol{z_t} = [z_t^1, \ldots, z_t^n], \boldsymbol{a_t} = [a_t^1, \ldots, a_t^n]$. Although Eq.(1) includes an actor, it does not serve as an explicit decision-making policy. Instead, it is used solely to generate candidate actions for the planner.

**Learning Objectives.** Let $\theta_E$, $\theta_D$, $\theta_R$, $\psi$, and $\phi$ denote the parameters of the encoder, dynamics model, reward predictor, critic, and actor, respectively. The world model are optimized under a self-supervised framework [10, 13, 1], and the loss function is defined as:

$$
\mathcal{L}(\theta, \psi) = \sum_i^n \sum_t^H \lambda^t \left( \underbrace{\left\| \hat{z}_t^i - z_{t+1}^i \right\|_2^2}_{\text{Dynamic Loss}} + \underbrace{SoftCE\left(\hat{r}_t, r_t\right)}_{\text{Reward Loss}} + \underbrace{SoftCE\left(\hat{q}_t, G_t\right)}_{\text{Q Loss}} \right),
\tag{2}
$$

where $\theta = \{\theta_E, \theta_D, \theta_R\}$, $H$ is the prediction horizon, $\lambda \in (0, 1]$ is a constant coefficient that balances the contribution of each rollout step, $G_t$ is the TD target, $z_{t+1}^i = \text{sg}\left(p_E(o_{t+1}^i)\right)$ is the encoded next latent state.

Following Eq. (2), the encoder learns a transformation from real observations to latent representations. The dynamics model and predictor are trained to minimize the prediction loss in latent space, ensuring alignment between predicted trajectories and real environment transitions.

## 4.2 Incorporate MoE in Multi-Agent World Model

According to the modeling in Section 4.1, the dynamics learning is formulated as a sequence-to-sequence prediction problem, while reward prediction is treated as a sequence-to-scalar regression problem. To align with these formulations, we implement the dynamics model using SoftMoE [28] and the reward predictor using SparseMoE [30].

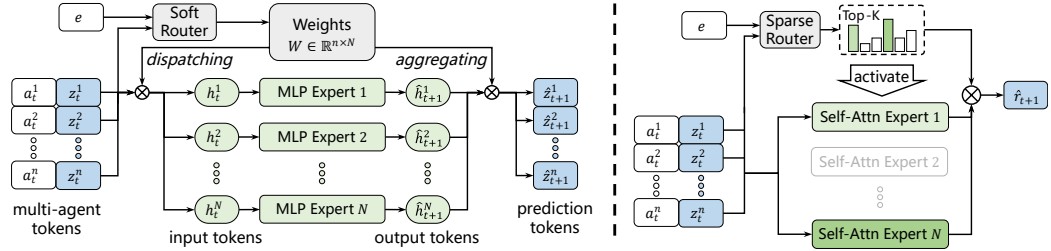

Figure 3: MoE in world model. The router generates a set of weights based on the task at hand, which is used for activating and allocating the experts. (*Left*) SoftMoE-based multi-agent dynamics model. (*Right*) SparseMoE-based reward predictor.

**SoftMoE for Multi-Agent Dynamics Learning.** In sequence $[(z_t^1, a_t^1), \ldots, (z_t^n, a_t^n)] \in \mathbb{R}^{n \times (d_{\mathcal{Z}} + d_{\mathcal{A}})}$, each pair $(z_t^i, a_t^i)$ can be treated as a token. The dynamics model, shown on the left side of Figure 3, is implemented as a variant of SoftMoE [28], comprising a soft router and a set of $N$ experts. The soft router computes a routing weight matrix as:

$$
W = \text{SoftRouter}(\boldsymbol{z_t}, \boldsymbol{a_t}, e) \in \mathbb{R}^{n \times N},
\tag{3}
$$

which plays two roles: (1) *dispatching* $n$ multi-agent tokens $(z_t^i, a_t^i)$ to $N$ input tokens $h_t^k$ for the experts; (2) *aggregating* the $N$ experts' output tokens $\hat{h}_{t+1}^k$ into $n$ predicted latent tokens $\hat{z}_{t+1}^i$. Here, $i = 1, \ldots, n$ and $k = 1, \ldots, N$.

For the first role, at the input stage, we apply `softmax` normalization along the rows of $W$ and compute $N$ input tokens as weighted combinations of the $n$ multi-agent tokens:

$$
h_t^k = \sum_{i=1}^n \frac{\exp(W_{ki})}{\sum_j \exp(W_{kj})} (z_t^i \oplus a_t^i) \in \mathbb{R}^{d_{\mathcal{Z}} + d_{\mathcal{A}}}, \quad k = 1, \ldots, N,
\tag{4}
$$

where $\oplus$ denotes the concatenation operation. Each $h_t^k$ serves as the input to the $k$-th expert, implemented as an independent MLP to avoid gradient interference. The experts produce $N$ output tokens $\hat{h}_{t+1}^k = \text{Expert}^k(h_t^k) \in \mathbb{R}^{d_z}$.

For the second role, at the output stage, we apply `softmax` normalization along the columns of $W$ to combine $N$ outputs of the experts and yield the final prediction $[\hat{z}_{t+1}^1, \ldots, \hat{z}_{t+1}^n]$:

$$\hat{z}_{t+1}^i = \sum_{k=1}^{N} \frac{\exp(W_{ki})}{\sum_j \exp(W_{ji})} \hat{h}_{t+1}^k \in \mathbb{R}^{d_z}, \quad i = 1, \ldots, n. \tag{5}$$

Compared to existing works that directly use MLPs or Transformers for multi-agent dynamics learning [7, 1, 45], our proposed dynamics model offers better adaptability to multi-task context. The set of experts can be viewed as a knowledge library, while the router can invoke and allocate experts based on the task at hand, effectively leveraging the property of **bounded similarity**. For *similarity* aspect, the router assigns similar experts to similar tasks, enhancing knowledge sharing and learning efficiency; for *bounded* aspect, it assigns different experts to dissimilar tasks, isolating gradients and avoiding conflicts.

**SparseMoE for Reward Prediction.** Unlike multi-agent dynamics, the reward is shared among all agents, making reward prediction a sequence-to-scalar regression problem. As shown on the right side of Figure 3, the reward predictor is implemented as a SparseMoE, consisting of a sparse router and a set of self-attention experts.

The sparse router generates a length-$N$ vector, representing the relevance scores of $N$ experts for the current task. During training, this vector is transformed into a probability distribution via `softmax`, and $K$ experts are selected via roulette sampling. During execution, the top-$K$ experts with the highest scores are directly activated. This allows the sparse router to invoke the most suitable experts based on the task at hand. Each expert in the predictor consists of a multi-head attention layer followed by an average pooling layer. The attention mechanism captures inter-agent interactions, while the pooling layer produces a unified reward prediction shared by all agents.

**Load-Balance Loss.** To mitigate the load imbalance issue in SparseMoE, some experts are overloaded with tasks while others remain inactive, we incorporate a load-balancing loss in [8].

$$\mathcal{L}_P(\theta_R) = \sum_{j=1}^{N} f_j P_j, \quad f_j = \frac{N}{K|\mathcal{T}|} \sum_{\mathcal{M} \in \mathcal{T}} \mathbb{1}(\mathcal{M} \text{ selects Expert } j), \quad P_j = \frac{1}{|\mathcal{T}|} \sum_{\mathcal{M} \in \mathcal{T}} s_j(\mathcal{M}), \tag{6}$$

where $f_j$ denotes the selection frequency of expert $j$, $P_j$ represents its average task-relevance score, and $s_j(\mathcal{M})$ is the score of expert $j$ for task $\mathcal{M}$.

### 4.3 Learning and Planning Multi-Agent Tasks

We then integrate the MoE-based world model with an actor-critic module to enable learning and planning in multi-agent tasks. A key advantage of M3W's planning is that it leverages the predictive capability of the world model directly, without relying explicitly on a decision-making policy, thereby avoiding the limitations of policy-centric methods. Action planning in M3W is realized using a multi-agent extension of the model predictive path integral (MPPI) algorithm [39]. At timestep $t$, given an action sequence $\boldsymbol{a}_{t:t+H}$, the world model performs rollouts in the latent space to generate predicted rewards $r_{t:t+H}$ and a latent terminal state $\boldsymbol{z}_{t+H}$. By combining these predictions with the critic's estimate of the Q-value at the terminal state, the expected return of each action sequence can be evaluated as:

$$V_{\boldsymbol{a}_{t:t+H}} = \gamma^H Q\left(\boldsymbol{z}_{t+H}, \boldsymbol{a}_{t+H}, e\right) + \sum_{h=0}^{H-1} \gamma^h p_R\left(\boldsymbol{z}_{t+h}, \boldsymbol{a}_{t+h}, e\right), \tag{7}$$

where the first term represents the long-term return, estimated by the critic, and the second term represents the short-term cumulative rewards predicted by the world model. Then the policy of planner can be expressed as $\boldsymbol{\pi}^{\text{Planner}} = \arg\max_{\boldsymbol{a}_{t:t+H}} V_{\boldsymbol{a}_{t:t+H}}$.

As shown in Eq. (1), the world model of M3W includes an actor module. The actor does not serve as the explicit decision-making policy, but rather acts as a candidate action generator for the planner.

We adopt the Heterogeneous Multi-Agent Soft Actor-Critic (HASAC) algorithm[20] to optimize the actor, with the loss function defined as:

$$\mathcal{L}(\phi) = \sum_i^n \sum_t^H \lambda^t \left( \alpha \mathcal{H} \left( \pi^{\text{Actor}} \left( \cdot \left| z_t^i, e \right. \right) \right) - Q \left( \boldsymbol{z_t}, \boldsymbol{a_t}, e \right) \right), \tag{8}$$

where $\alpha$ is the entropy coefficient, $\mathcal{H}(\cdot)$ denotes the entropy function. For the second term, $\boldsymbol{z_t} = [z_t^1, \ldots, z_t^n]$ and $z_0^i = p_E \left( o_0^i, e \right)$ is the encoded initial latent state, $z_t^i = q_D \left( \boldsymbol{z_t}, \boldsymbol{a_t}, e \right)^i$ is the predicted states from dynamics model, $\boldsymbol{a_t} = [a_t^1, \ldots, a_t^n]$ and $a_t^i = \pi^{\text{Actor}} \left( z_t^i, e \right)$.

Details of the multi-agent planner, pseudocode for learning and planning, and implementation specifics are provided in Appendix A, B, and C, respectively. Further discussion and ablation studies on the role of the actor module appears in Appendix E.4.

## 5 Experiments

In this section, we evaluate the following properties of our method: (a) performance in multi-task learning, particularly in comparison with state-of-the-art policy-centric methods; (b) effectiveness of the MoE-based world model, assessing whether the MoE improves prediction accuracy and planning quality; (c) interpretability, including M3W's exploitation of bounded similarity, expert specialization, and the diversity of multi-task behaviors.

**Environments.** We evaluate M3W and the baselines on on two challenging benchmarks: Bimanual Dexterous Hands (Bi-DexHands)[4] with 14 tasks, and the multi-agent Mujoco (MA-Mujoco)[26] with 24 tasks. Both benchmarks include a mix of similar and diverse tasks, providing a solid testbed for multi-task performance. In Bi-DexHands, two agents control a pair of dexterous hands to perform complex manipulations, with high-dimensional observation and action spaces (up to $\mathcal{O} \in \mathbb{R}^{229}, \mathcal{A} \in \mathbb{R}^{26}$). In MA-MuJoCo, agents coordinate by controlling different joints of a MuJoCo robot to accomplish locomotion and control tasks. More detailed descriptions, visualizations and comparisons with the other environments can be found in Appendix D.

### 5.1 Comparisons

**Baselines.** For the baselines, MACD [1] introduces a transformer-based world model, serving as a baseline for model-based methods. HMASD [40] applies modularity to hierarchical skill discovery and serves as a baseline for multi-task methods. MAT [38], MAPPO [41], and HATRPO [19] are among the most popular general MARL algorithms currently available. Additionally, we trained a separate MAPPO model for each task as a strong baseline.

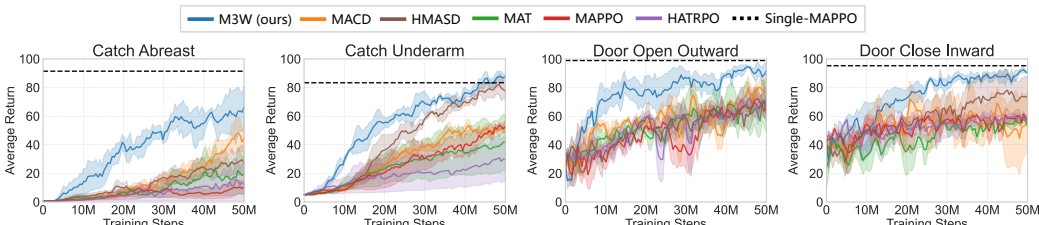

Figure 4: Performance comparisons for selected tasks in Bi-DexHands. The black dashed line represents the strong baseline under single-task MAPPO, while the remaining bold curves indicate the average returns, with the shaded areas representing the 95% confidence intervals.

**Results on Bi-DexHands.** Figure 4 shows the performance curves of M3W and baselines on the selected tasks in Bi-DexHands, while Table 1 reports results across all tasks. Overall, M3W achieves superior performance, better sample efficiency, and greater multi-task adaptability. After 50M steps, M3W consistently outperforms the baselines on average and closely matches the strong baseline on most individual tasks.

For sample efficiency, consider several complex yet similar tasks, such as `catch-abreast`, and `catch-underarm`. These tasks all involve object throwing and catching between two hands, sharing

Table 1: Performance comparisons for various tasks in Bi-DexHands. The values following the ± indicate the 95% error margins. The maximum values are highlighted in bold with blue shading. The rewards are normalized to the range [0, 100], and the original rewards and normalization methods can be found in Appendix E.1.

| Task | M3W(ours) | MACD [1] | HMASD [40] | MAT [38] | MAPPO [41] | HATRPO [19] |
|------|-----------|----------|------------|----------|------------|-------------|
| over | **70.0 ± 1.3** | 64.4 ± 0.9 | 62.3 ± 0.8 | 54.6 ± 1.6 | 63.1 ± 1.0 | 42.3 ± 4.3 |
| bottle-cap | 89.8 ± 0.6 | 85.7 ± 1.1 | 90.1 ± 0.6 | 89.6 ± 0.4 | **90.9 ± 0.5** | 88.2 ± 0.7 |
| catch-abreast | **64.1 ± 4.4** | 44.5 ± 3.1 | 28.6 ± 3.0 | 19.1 ± 4.2 | 10.0 ± 1.5 | 13.0 ± 3.8 |
| catch-over2underam | **84.8 ± 1.4** | 79.9 ± 2.0 | 79.5 ± 3.9 | 42.9 ± 3.8 | 69.7 ± 2.1 | 70.3 ± 4.6 |
| catch-underarm | **87.3 ± 1.3** | 51.7 ± 1.4 | 78.8 ± 2.2 | 41.4 ± 5.5 | 52.1 ± 1.3 | 29.2 ± 4.4 |
| block-stack | **88.1 ± 3.1** | 72.5 ± 2.4 | 60.5 ± 7.3 | 62.5 ± 1.3 | 59.5 ± 3.3 | 62.5 ± 3.1 |
| door-close-inward | **91.3 ± 0.7** | 54.0 ± 6.0 | 73.5 ± 4.3 | 56.4 ± 0.8 | 59.3 ± 0.9 | 58.4 ± 0.8 |
| door-close-outward | **95.0 ± 1.2** | 92.4 ± 1.0 | 85.6 ± 2.5 | 82.8 ± 2.0 | 74.8 ± 3.1 | 86.5 ± 0.5 |
| door-open-inward | 91.5 ± 0.4 | **92.7 ± 0.8** | 82.5 ± 1.3 | 83.7 ± 1.1 | 82.2 ± 1.2 | 80.6 ± 1.7 |
| door-open-outward | **89.0 ± 2.1** | 77.5 ± 2.5 | 68.9 ± 3.1 | 68.6 ± 4.6 | 65.8 ± 1.6 | 69.4 ± 2.2 |
| kettle | **90.4 ± 0.4** | 86.1 ± 0.9 | 89.9 ± 1.1 | 87.8 ± 0.7 | 84.1 ± 1.0 | 86.2 ± 1.3 |
| lift-underarm | **77.0 ± 2.1** | 60.8 ± 3.3 | 57.7 ± 2.1 | 46.6 ± 2.7 | 40.5 ± 1.7 | 46.6 ± 3.2 |
| pen | **83.3 ± 3.3** | 69.6 ± 3.5 | 68.6 ± 2.7 | 29.7 ± 6.6 | 58.5 ± 7.5 | 29.9 ± 2.9 |
| scissors | 76.4 ± 2.3 | 76.7 ± 4.3 | **86.0 ± 6.8** | 57.7 ± 4.7 | 80.1 ± 3.3 | 75.2 ± 2.1 |
| average | **84.1 ± 0.8** | 72.0 ± 1.2 | 72.3 ± 1.4 | 58.8 ± 1.6 | 63.6 ± 1.4 | 59.4 ± 1.6 |

the same observation/action spaces as well as similar dynamics. As shown in Figure 4, M3W achieves significantly faster convergence and even surpasses the strong single-task baseline in the latter task. This demonstrates that the MoE-based world model effectively exploits task similarity, enabling rapid acquisition of similar dynamics and improved sample efficiency.

For adaptability in multi-task learning, taking `door-open-outward` and `door-close-inward` as examples, they share the same observation/action spaces and dynamics, yet have completely disparate initial states and reward functions. As shown in the last two plots of Figure 4, most baselines struggle to perform well on both tasks concurrently—e.g., HMASD performs slightly better on `door-close-inward` but still learning slowly `door-open-outward`. In contrast, M3W successfully learns both opposing tasks simultaneously. This flexibility is attributed to the modular structure of the world model, which effectively mitigates semantic conflicts between tasks.

**Results on MA-Mujoco.** The performance comparison of the selected tasks in MA-Mujoco can be found in Figure 5, and the complete results in MA-Mujoco can be found in Appendix E.1. The tasks in MA-Mujoco involves multiple robotic suites, each corresponding to multiple tasks. Within the same suite, the tasks share dynamics while differing in their reward functions. While across different suites, the dynamics are also distinct, which is a challenge for the model to effectively knowledge reuse.

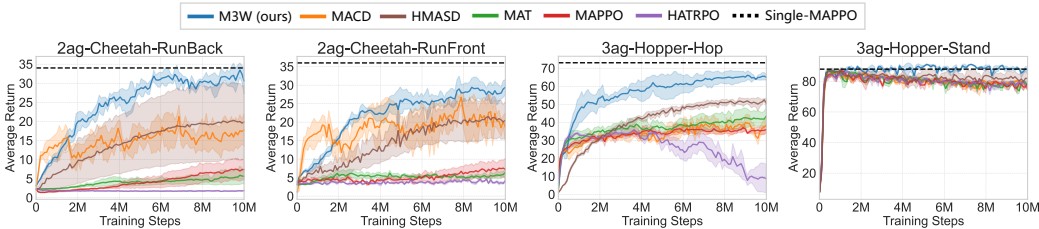

Figure 5: Performance comparisons for selected tasks in MA-Mujoco.

Figure 5 (left) presents results on two tasks from the `cheetah` suite, including `run-front` and `run-back`, which encourage the robot to run using only its front-leg or back-leg, respectively. These tasks share the same dynamics, and M3W achieves higher sample efficiency, indicating that the world model effectively enables knowledge sharing across similar tasks. Figure 5 (right) shows the tasks `hopper-hop` and `hopper-stand`, which involve robots that are entirely different from the `cheetah`. In these tasks, M3W also demonstrates superior performance. Notably, for simple tasks like `hopper-stand`, only M3W maintains consistent performance post-convergence, whereas other baselines suffer degradation due to interference from other tasks. This further confirms the advantages of M3W in addressing multi-task knowledge conflicts.

For statistical analysis of the comparison results, please refer to Appendix E.2.

## 5.2 Effectiveness of MoE-based World Model

We then conduct experiments to evaluate the accuracy of the MoE-based world model and its contribution to action planning, providing both direct and indirect evidence of its effectiveness.

**Accuracy of the World Model Prediction**. We selected six tasks—{`Cheetah: run, run-bwd, run-front; Walker2D: run, run-bwd, walk`}—involving two Mujoco robots whose dynamics exhibit a clear pattern of bounded similarity. We replace MoE with other alternative architectures, such as Transformer, Switch-Transformer and MLP, then retrain the models, and evaluate their prediction accuracy. Results are reported in Figure 6. Compared to other structures, M3W achieves the lowest prediction error and highest stability, along with strong long-range prediction capability. More detailed experimental settings and models' parameter counts can be found in Appendix E.3.

For dynamics prediction, *MLP* performs the worst, because the local dynamics is non-stationary and difficult to fit with a shared model. *Transformer* [1, 45] uses an attention mechanism to model interactions between agents, learning dynamics from a global perspective. While it shows some improvement over *MLP*, a single set of shared parameters still limits its adaptability across multiple tasks. *Switch-Transformer* [8] replaces the feed-forward network (FFN) with MoE, but due to the lack of proper dispatching and aggregation mechanisms, each expert operates only on local observations and actions. As a result, it also struggles to capture diverse dynamics. For reward prediction, *MLP* again exhibits poor performance. *Self-Attention* also suffers from limited generalization due to parameter sharing. Only our method *M3W* achieves the highest accuracy and stability across tasks.

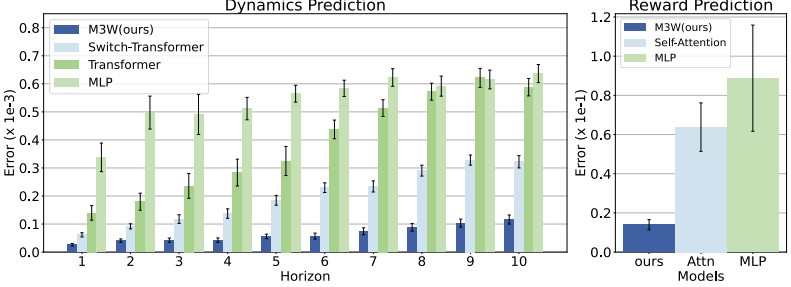

Figure 6: Accuracy of the world model prediction. (*Left*) Prediction errors on dynamics, where the height of each bar indicates the average error, and the horizontal lines represent the 95% confidence intervals. (*Right*) Prediction errors on rewards.

**Contribution to Model-based Planner.** The performance of the planner serves as an indirect indicator of the effectiveness of the world model. As discussed in Section 4.3, the planner does not involve any learnable parameters—its performance relies entirely on the quality of predicted trajectories. The more accurate the predictions, the greater the performance boost from planning. To further examine this, we equip MACD with the same planner, and the results are presented in Figure 7. We observe that adding the planner does lead to performance improvements for MACD, but a significant gap remains between it and M3W. This further supports, from an indirect perspective, the accuracy and effectiveness of our MoE-based world model in generating reliable multi-task predictions.

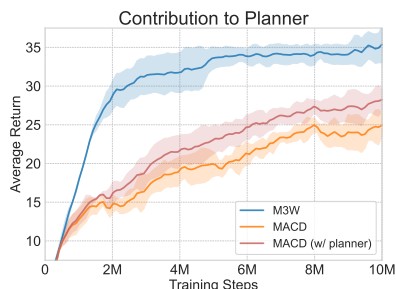

Figure 7: Performance impact of adding a planner to MACD.

## 5.3 Interpretability and Visualization

In the following, we conduct a series of visualization experiments to illustrate the interpretability of M3W.

**Expert Specialization.** We visualize the attention distributions of individual experts during the `Walker-run` task in Figure 8. The results reveal that the highest-scoring expert consistently focuses on the front leg in contact with the ground. It is a reasonable pattern, as these joints are actively

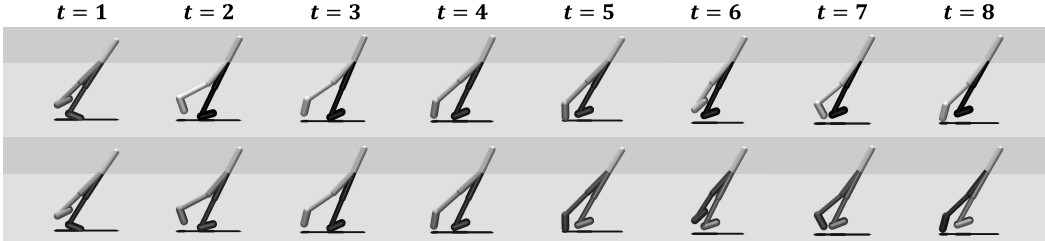

Figure 8: Visualization of the attention outputs of experts in the reward predictor on robot joints, where deeper colors indicate higher weights. (*Top*) Highest-scoring expert; (*Bottom*) Lowest-scoring expert.

exerting force and play a critical role in determining the robot's movement. In contrast, the lowest-scoring expert exhibits an implausible attention pattern, maintaining focus on the same rear leg across multiple time steps (e.g., at $t = 5, 6, 7, 8$), without adapting to the movement. This contrast highlights that the experts in M3W have achieved functional **specialization**, and that the router has successfully learned to assign the appropriate expert to task at hand.

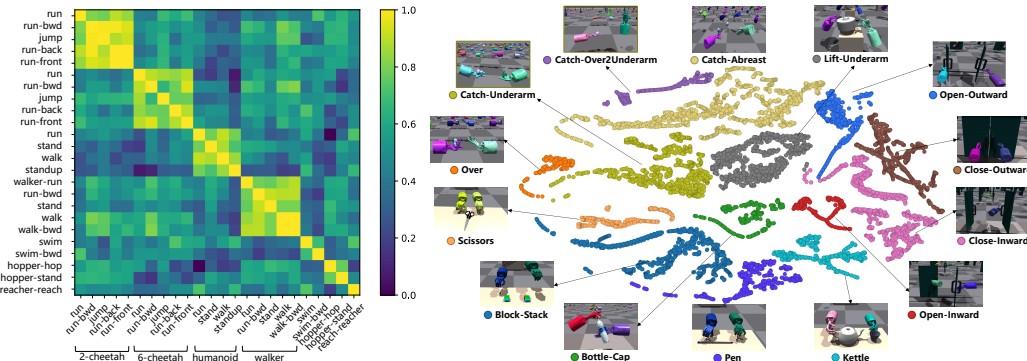

Figure 9: (*Left*) Visualization of cosine similarity of router outputs. (*Right*) t-SNE trajectories of the joint latent states.

**Exploiting Bounded Similarity.** As discussed in Section 4.2, the experts set can be viewed as a knowledge library, with the router invoking and allocating them based on the task at hand. To better understand this mechanism, we visualized the pairwise cosine similarity of router outputs across tasks, as shown on the left of Figure 9. The resulting pattern exhibits clear diagonal blocks along the main diagonal, which correspond to tasks with similar dynamics. This indicates that the SoftMoE-based dynamics model successfully leverages the bounded similarity among tasks, achieving expert reuse and effective generalization across related tasks.

**Multi-Task Behavior Diversity**. The right side of Figure 9 visualizes multi-task trajectories in the shared latent space. The results show clearly separated regions for different tasks, indicating that the model captures diverse behaviors across tasks. Furthermore, tasks with similar dynamics tend to cluster together—for instance, `catch-abreast`, `catch-over2underarm`, and `catch-underarm` are positioned closely, as are the `close/open-inward/outward` tasks. This observation aligns with M3W's design motivation and supports its effectiveness in multi-task learning.

## 5.4   Other Experimental Analysis

To further examine the characteristics of M3W, we present the following additional analyses:

1) The impact of the actor prior in Appendix E.4;
2) The robustness of M3W to variations in reward scale in Appendix E.5;
3) The inference time cost analysis in Appendix E.6;
4) The scalability evaluation with respect to the number of agents in Appendix E.7;

# 6 Conclusion

In this paper, we propose M3W, which leverages a MoE-based world model for learning and planning multi-agent tasks. Motivated by the finding of bounded similarity, we reconsider the role of modularity in multi-task learning and propose to integrate it into the world model rather than the policy network. Specifically, a SoftMoE-based dynamics model and a SparseMoE-based reward predictor are used to isolate dissimilar task dynamics while promoting knowledge reuse among similar tasks. By directly performing action planning on synthetic trajectories generated by the world model, M3W overcomes the limitations of policy-centric methods. Experimental results on Bi-DexHands and MA-Mujoco demonstrate that M3W significantly improves both performance and sample efficiency. Comprehensive ablation studies and visualizations further confirm the critical role of the MoE-based architecture in capturing bounded similarity and enabling effective multi-task planning.

Currently, M3W is only applicable to tasks with continuous action spaces, due to the limitations of the MPPI-based planner. Future work could integrate M3W with Monte Carlo Tree Search (MCTS) or Cross-Entropy Method (CEM)-based planner to accommodate tasks in discrete action spaces.

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

# Appendices

# A   Multi-agent Planner

We provide a detailed implementation of the multi-agent planner mentioned in Section 4.3. The planning process is illustrated in Figure 10.

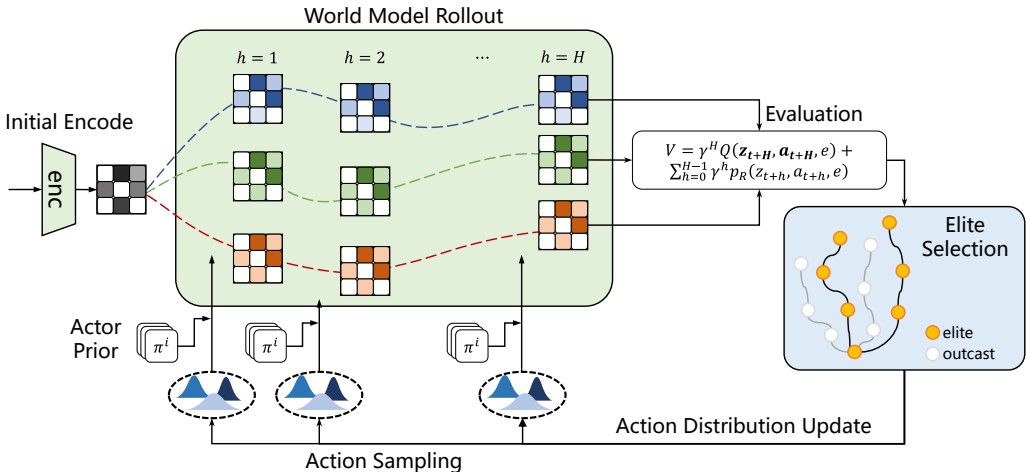

Figure 10: The process of model-based planning.

**Action Sampling.** At each decision timestep $t$, a diagonal Gaussian distribution is used to represent the action distribution for each agent, and $N_p$ action sequences are sampled, $a^i_{t:t+H} \sim \mathcal{N}\left(\mu^i_{t:t+H}, (\sigma^i_{t:t+H})^2 I\right)$, where $\mu^i_{t:t+H} \in \mathbb{R}^{|\mathcal{A}| \times H}$, $\sigma^i_{t:t+H} \in \mathbb{R}^{|\mathcal{A}| \times H}$ are the mean and standard deviation of the action distribution for agent $i$.

**Actor Prior.** We incorporate $N_\pi < N_p$ action sequences with policy priors into the candidate set, which are obtained directly from the actor. This inclusion aids in accelerating the convergence of the planner during the early iterations, while ensuring the foundational performance of the model-based planner.

**World Model Rollout.** After action sampling, the world model of M3W performs $H$-step rollouts in the latent space, resulting in $N_p$ predicted trajectories of length $H$, one of which is denoted as $\Gamma_m = \left\{ \left[\hat{z}^1_\tau, \ldots, \hat{z}^n_\tau\right], \left[a^1_\tau, \ldots, a^n_\tau\right], \hat{r}_\tau \right\}_{\tau=t:t+H}$.

**Evaluation.** Subsequently, these trajectories are evaluated based on the value function using $H$-step return. Specifically, the value of trajectory $\Gamma_m$ is given by Eq. (7).

**Elite Selection.** Based on the estimated value estimated, the sampled trajectories are ranked, and the top $M$ trajectories are selected as elite set, one of which is denoted as $\Gamma^*_m$.

**Action Distribution Update.** The process of updating the action distribution can be formally expressed as Eq. (9), in which $\alpha_m = \exp\left[\tau\left(V_{\Gamma^*_m} - \max_{m \in M} V_{\Gamma^*_m}\right)\right]$ denotes the weight of each elite trajectory, and $\tau$ is a temperature coefficient.

$$\left(\mu^i_{t:t+H}\right)' = \frac{\sum_{m=1}^{M} \alpha_m \Gamma^*_m}{\sum_{m=1}^{M} \alpha_m}, \quad \left(\sigma^i_{t:t+H}\right)' = \sqrt{\frac{\sum_{m=1}^{M} \alpha_m \left(\Gamma^*_m - \left(\mu^i_{t:t+H}\right)'\right)^2}{\sum_{m=1}^{M} \alpha_m}}, \quad (9)$$

**Iteration.** The above process is iterated $K_p$ times to derive the final action distribution.

The detailed hyperparameters used in the model-based planner are summarized in Table 2.

Table 2: The Notations and Values of hyperparameters in the planner.

| Hyperparameters | Notations | Value | Hyperparameters | Notations | Value |
|---|---|---|---|---|---|
| rollout horizon | $H$ | 3 | sampling actions | $N_p$ | 512 |
| planning iterations | $K_p$ | 6 | elites | $M$ | 64 |
| temperature | $\tau$ | 0.5 | actor prior samples | $N_\pi$ | 24 |

## B  Pseudocode

We present the pseudocode for M3W training and planning, as shown in Algorithm 1 and Algorithm 2, respectively.

---

**Algorithm 1** Model Training

---

**Input:** Task set $\mathcal{T} = \{\mathcal{M}_k\}$, replay buffer $\mathcal{B}$, task embeddings $e_k$; parameterized networks $\phi, \psi$ for actor and critic function, $\theta_E, \theta_D, \theta_R$ for encoder, dynamics model and reward predictor
**for** episode $= 1, 2, 3, \ldots,$ **do**
    **for** step $t = 1, 2, 3, \ldots$ **do**
        Get real data $([o_t^i]_{i=1:n}, [a_t^i]_{i=1:n}, r_t, [o_{t+1}^i]_{i=1:n})$ by interacting with the environment
        Add transition into buffer: $\mathcal{B} = \mathcal{B} \cup ([o_t^i]_{i=1:n}, [a_t^i]_{i=1:n}, r_t, [o_{t+1}^i]_{i=1:n})$
    **end for**
    **for** epoch $= 1, 2, 3, \ldots,$ **do**
        **for** task $\mathcal{M}_k \in \mathcal{T}$ **do**
            Sample trajectories from $\mathcal{B}$
            Update $\theta_E, \theta_D, \theta_P, \psi$ by minimizing Eq. (2) & Eq. (6)
            Update $\phi$ by minimizing Eq. (8).
        **end for**
    **end for**
**end for**

---

**Algorithm 2** Model Planning

---

**Input:** learned parameters $\theta_E, \theta_D, \theta_P, \phi, \psi, e$, hyperparameters $H, K_p, \tau, N_p, M, N_\pi$, initial distribution;
**for** step $t = 1, 2, 3, \ldots$ **do**
    Get environment observation $[o_t^i]_{i=1:n}$ and encode them to latent space: $\boldsymbol{z_t} = p_E(\boldsymbol{o_t}, e; \theta_E)$
    **for** iteration $= 1, 2, 3, \ldots, K_p$ **do**
        Sample $N_p$ actions $a_{t:t+H}^i \sim \mathcal{N}\left(\mu_{t:t+H}^i, (\sigma_{t:t+H}^i)^2 I\right)$
        Sample $N_\pi$ actions from actor $\pi^i(z_t^i, e, i; \phi)$
        Get predicted data by world model rollouts, $\Gamma_m = \left\{\left[\hat{z}_\tau^1, \ldots, \hat{z}_\tau^n\right], \left[a_\tau^1, \ldots, a_\tau^n\right], \hat{r}_\tau\right\}_{\tau=t:t+H}$
        Evaluate the trajectories by Eq. (7) and select $M$ elite action sequences $\Gamma_m^* = \arg\max V_{\Gamma_m}$
        Update action distribution following Eq. (9)
    **end for**
**end for**

---

## C  Implement Details

### C.1  Discrete Regression of Reward and Value

We discretize the reward predictions and value estimates, transforming the scalar regression into a discrete regression to enhance robustness [12, 13, 33]. Taking reward prediction as an example, we partition the range of rewards into $B$ bins, and the scalar reward can be transformed into a $B$-dimensional two-hot vector. The two-hot encoding process is defined as follows: for a given scalar value $r$, we locate the adjacent bins $b_k \leq r < b_{k+1}$, and then assign weights to the two corresponding

positions based on linear interpolation. Formally, the encoding is defined as:

$$\text{two-hot}(r)_d = \begin{cases} |b_{k+1} - r|/|b_{k+1} - b_k|, & \text{if } d = k \\ |b_k - r|/|b_{k+1} - b_k|, & \text{if } d = k + 1 \\ 0, & \text{else} \end{cases}, k = \sum_{j=1}^{B} \mathbb{1}(b_j \leq r). \tag{10}$$

During training, we used the soft cross-entropy loss to optimize the discretized reward and value estimation. For Eq. (2), its last two loss terms are replaced as follows:

$$(q_R(\boldsymbol{z_t}, \boldsymbol{a_t}, e) - r_t)^2 \leftarrow \sum_{d}^{D} \frac{\exp(q_R(\boldsymbol{z_t}, \boldsymbol{a_t}, e)_d) \times \text{two-hot}(r_t)_d}{\sum_d \exp(q_R(\boldsymbol{z_t}, \boldsymbol{a_t}, e)_d)} \tag{11}$$

## C.2 Symlog Predictions

To further enhance numerical stability, we apply the symmetric logarithmic (symlog) and exponential (symexp) transformations. These transformations are particularly effective in scenarios where target values exhibit high variance, such as in reward prediction or value estimation. They preserve continuity and differentiability, enabling stable learning while maintaining meaningful relationships between values. These transformations are defined in [12] as:

$$\text{symlog}(x) = \text{sign}(x)\ln(|x| + 1), \quad \text{symexp}(x) = \text{sign}(x)(\exp(|x|) - 1). \tag{12}$$

## C.3 Percentile Scaling

We adopt a percentile-based scaling method [13]. This method dynamically adjusts the scaling of Q-values by calculating the difference between the 5th and 95th percentiles of the returns in each batch. The scaling factor is updated smoothly to maintain numerical stability.

## C.4 SimNorm Normalization

We adopt SimNorm (Simplicial Normalization) [13] to construct the latent space. It projects the latent state onto a simplex of fixed dimensionality using a softmax operation. SimNorm is particularly effective in high-dimensional spaces, as it promotes sparse representations, mitigates information loss or degradation, and preserves the structural integrity of the learned representation. The SimNorm transformation is defined as:

$$\text{SimNorm}(x) = [g_1, \ldots, g_N], \quad , g_i = \frac{\exp(x_{i:i+L}/\tau)}{\sum^{L} \exp(x_{i:i+L}/\tau)}, \tag{13}$$

where $L$ is the simplex dimension, $\tau$ is the temperature parameter, $N = d_x/L$ is the number of simplex segments, and $x_{i:i+L}$ is the $i$-th segment of the input vector.

## C.5 Hyperparameters

The hyperparameters for the model-based planner have been provided in Appendix A. Here, we present the hyperparameters regarding the the training and network structures in Table 3 and Table 4.

Table 3: The training hyperparameters.

| Hyperparameters | Value | Hyperparameters | Value | Hyperparameters | Value |
|---|---|---|---|---|---|
| buffer size | 1e6 | batch size | 256 | train interval | 1 |
| step balance $\lambda$ | 0.5 | lr | 5e-4 | encoder lr | 1.5e-4 |
| n-step return | 10 | gamma | 0.99 | | |

# D Additional Description of Environments

## D.1 Description of Bi-DexHands tasks

Bimanual Dexterous Hands (Bi-DexHands) [4] is a widely adapted cooperative multi-agent environment, in which two agents control two ShadowHands to perform dexterous manipulation tasks.

Table 4: The network configurations.

| Hyperparameters | Value | Hyperparameters | Value | Hyperparameters | Value |
|---|---|---|---|---|---|
| task dim | 96 | latent dim | 512 | encoder size | [256] |
| SimNorm dim | 8 | num experts | 16 | experts size | [512, 512] |
| predictor $K$ | 2 | actor & critic size | [512, 512] | num bins | 101 |
| scale $\rho$ | 0.01 | dynamics coef | 20 | reward coef | 0.1 |
| q coef | 0.1 | entropy coef | 0.01 | | |

We selected 14 representative tasks that comprise the Bi-DexHands task set, which are visualized on the top of Figure 11. Bi-DexHands features a high-dimensional observation and action space, up to $\mathcal{O} \in \mathbb{R}^{229}, \mathcal{A} \in \mathbb{R}^{26}$, respectively. Table 5 presents a list of tasks in Bi-DexHands, including the dimensions of the observation and action spaces, along with a brief description of each task.

Table 5: Task descriptions in Bi-DexHands.

| $n \times d_{\mathcal{O}} \times d_{\mathcal{A}}$ | Task | Description |
|---|---|---|
| $2 \times 211 \times 20$ | Over | Two palm-up hands oriented oppositely transfer an object; one hand holds and throws the object while the other hand catches it (the degrees of freedom at the hand's base are frozen). |
| $2 \times 221 \times 26$ | Bottle-Cap | Two hands collaborate to open the bottle cap; one hand holds the bottle while the other hand removes the cap. |
| $2 \times 229 \times 26$ | Block-Stack | Two hands each pick up one block and stack them together. |
| $2 \times 223 \times 26$ | Catch-Abreast | Two palm-up hands oriented in the same direction pass an object; one hand holds the object while the other hand catches it. |
| | Over2Underarm | One vertical hand throws an object while the other palm-up hand catches it. |
| | Catch-Underarm | Two palm-up hands oriented oppositely transfer an object; one hand holds the object while the other hand catches it (the degrees of freedom at the hand's base are not frozen). |
| $2 \times 218 \times 26$ | Close-Inward | Two hands close a door that opens outward by pulling it inward. |
| | Close-Outward | Two hands close a door that opens inward by pushing it outward. |
| | Open-Inward | Two hands open a door by pulling it inward. |
| | Open-Outward | Two hands open a door by pushing it outward. |
| | Kettle | Two hands pouring water collaboratively, one hand lifts the kettle and pours water into a bucket held by the other hand. |
| | Lift-Underarm | Two hands grasp the handle of a pot and lift it to a specified position. |
| | Pen | Two hands cooperate to open the cap from a pen. |
| | Scissors | Two hands work together to open a pair of scissors. |

## D.2 Description of MA-Mujoco tasks

Multi-agent Mujoco (MA-Mujoco) [26] is an extended version of Mujoco designed for multi-agent cooperation. It divides a Mujoco suite into multiple agents based on joints and parts, allowing agents to collaboratively control robots to complete tasks. In this context, we selected 24 tasks to form the MA-Mujoco task set, with some of these tasks being adapted and expanded from DMControl [37]. These tasks exhibit a significant degree of shared dynamics, making them an ideal testing ground for evaluating knowledge transfer and reuse in multi-task learning algorithms. The visualizations of the tasks are presented on the bottom of Figure 11, and the descriptions of the tasks are provided in Table 6.

## D.3 Visualization of Tasks

## D.4 Why Choose Bi-DexHands and MA-Mujoco?

From the perspective of multi-task learning, we compare Bi-DexHands [4] and MA-Mujoco [26] with other multi-agent environments, including SMAC(StarCraft Multi-Agent Challenge) [29], MPE(Multi-Agent Particle Environment) [23], and RWARE(Multi-Robot Warehouse) [25], which are visualized in Figure 12.

We compare these environments from multiple aspects, as summarized in Table 7. It is evident that Bi-DexHands and MA-Mujoco are indeed challenging multi-task multi-agent environments.

Table 6: Task descriptions in MA-Mujoco.

| Suite | $n \times d_{\mathcal{O}} \times d_{\mathcal{A}}$ | Task | Description |
|---|---|---|---|
| Cheetah | $2 \times 7 \times 3$ | run | The front and back legs of the half-cheetah are treated as two agents that collaborate to control the robot to run forward. |
| | | run-bwd | The robot runs backwards |
| | | jump | Both legs leave the ground to jump while maintaining a lateral stationary position. |
| | | run-back | The robot runs forward using only the back legs. |
| | | run-front | The robot runs forward using only the front legs. |
| | $6 \times 4 \times 1$ | run | Each joint of the half-cheetah is treated as an agent, with the other settings being the same as those in 2ag-cheetah-run. |
| | | run-bwd | see 2ag-cheetah-run-bwd |
| | | jump | see 2ag-cheetah-jump |
| | | run-back | see 2ag-cheetah-run-back |
| | | run-font | see 2ag-cheetah-run-front |
| Humanoid | $2 \times 33 \times 9$ | run | The upper and lower halves of the humanoid are treated as two agents that collaborate to control the robot to run forward. |
| | | stand | The humanoid stands and maintains balance. |
| | | walk | The humanoid walks forward. |
| | | standup | The humanoid lying flat is controlled to stand up. |
| Walker2D | $2 \times 7 \times 3$ | run | The left and right legs of the walker2d are treated as two agents that collaborate to control the robot to run forward. |
| | | run-bwd | The walker2D runs backwards. |
| | | stand | The walker2D stands and maintains balance. |
| | | walk | The walker2D walks forward. |
| | | walk-bwd | The walker2D walks backwards. |
| Swimmer | $2 \times 8 \times 1$ | swim | The two joints of the swimmer are treated as two agents that collaborate to control the robot to swim forward. |
| | | swim-bwd | The robot swims backward. |
| Hopper | $3 \times 4 \times 1$ | hop | The three joints of the hopper are treated as three agents that collaborate to control the robot to hop forward. |
| | | stand | The Hopper stands and maintains balance. |
| Reacher | $2 \times 3 \times 1$ | reach | The two joints of the reacher are treated as two agents that collaborate to extend the robot's arm and touch the target. |

Table 7: Comparison of multi-task multi-agent environments.

| | Bi-DexHands | MA-Mujoco | SMAC | MPE | RWARE |
|---|---|---|---|---|---|
| **Semantics** | ✓ | ✓ | ✗ | ✓ | ✗ |
| **Diverse Dynamics** | ✓ | ✓ | ✗ | ✗ | ✗ |
| **Heterogeneous** | ✓ | ✓ | ✓ | ✗ | ✗ |

1. **Semantics.** In Bi-DexHands, tasks exhibit varying semantic relationships, including similar, opposite, and unrelated tasks. For instance, `catch-abreast` and `catch-underarm` are semantically similar, `close-inward` and `open-outward` are semantically opposite, while tasks like `pen` and `scissors` are entirely unrelated to others. A similar situation is observed in MA-Mujoco, where tasks such as *walker-run* and *walker-walk* are quite similar, and *walker-runbackwards* is opposite to them, while `reacher-reach` is entirely unrelated to others. In contrast, in SMAC and RWARE, the semantics of different tasks are fundamentally the same. For example, in SMAC, multi-task setting is primarily reflected in variations in agent types or numbers, but with the common semantic goal of "defeating the enemy to achieve victory." In RWARE, tasks differ in the layout of shelf positions, but share the common goal of "transporting goods to designated locations."

2. **Diverse Dynamics.** In Bi-DexHands, variations in manipulated objects and the freezing of certain degrees of freedom lead to changes in dynamics. For example, in the `over` tasks, the degrees of freedom at the hand's base are frozen, resulting in dynamics that differ from those of `catch` tasks. Similarly, in MA-Mujoco, different suites exhibit varying dynamics: *walker-run* and *walker-walk* share similar locomotion dynamics, but are unrelated to those in `reacher`. In contrast, the dynamics in other environments tend to be quite simple, where almost all tasks share identical dynamics. This simplicity limits their utility in assessing models' ability to generalize across dynamic variations.

3. **Heterogeneous.** Both Bi-DexHands and MA-Mujoco feature heterogeneous agents. In Bi-DexHands, two agents control the left and right hands separately, each with its own set

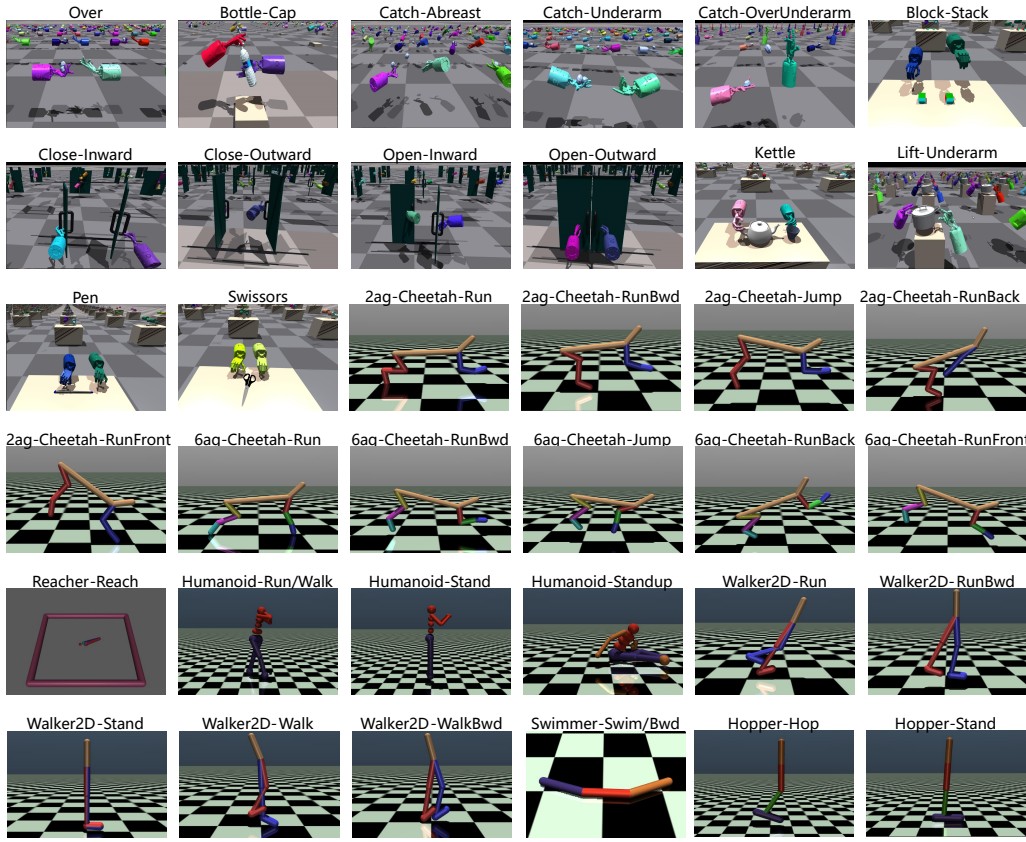

Figure 11: Visualization of tasks. (*Top*) Bi-DexHands tasks. (*Bottom*) MA-Mujoco tasks.

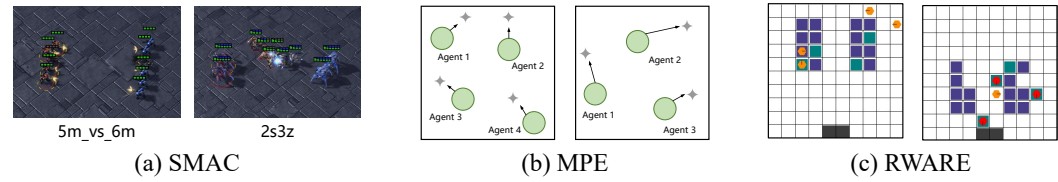

(a) SMAC            (b) MPE            (c) RWARE

Figure 12: Rendering of tasks in other multi-task multi-agent environments.

of observations and actions. In MA-Mujoco, different agents are responsible for controlling different joints of the robots, (e.g., hips vs. knees), leading to partial observability and coordination requirements. This agent-level heterogeneity poses a higher challenge for world modeling and multi-agent generalization.

Considering the above metrics, Bi-DexHands and MA-Mujoco exhibit diverse semantic relationships, varying dynamics, and significant agent heterogeneity, making them ideal for evaluating MT-MARL algorithms.

## D.5   Bounded Similarity in Other Environments

We believe that the *bounded similarity* phenomenon is not limited to continuous-control settings but represents a general property of multi-task environments. To further illustrate, we conducted experiments similar Figure 1 on other environments. Maximum Mean Discrepancy (MMD) [9] is used to measure pairwise similarity between task dynamics. The results are presented in Table 8, which demonstrate that tasks within the same group (e.g., MMM & MMM2) consistently exhibit much lower MMD scores than across-group pairs (e.g., 2s3z & MMM), indicating clear intra-group

similarity and inter-group divergence. This confirms that bounded similarity is consistently observed across diverse environments, validating its generality.

Table 8: Pairwise MMD scores among tasks in different environments (lower indicates higher similarity).

| SMAC | 2s3z | 3s5z | MMM | MMM2 |
|---|---|---|---|---|
| 2s3z | – | 0.15 | 0.58 | 0.69 |
| 3s5z | | – | 0.58 | 0.69 |
| MMM | | | – | 0.09 |
| MMM2 | | | | – |

| MPE | adversary | push | reference | spread |
|---|---|---|---|---|
| adversary | – | $< 0.01$ | 0.40 | 0.38 |
| push | | – | 0.39 | 0.38 |
| reference | | | – | 0.03 |
| spread | | | | – |

| RWARE | left-easy | left-hard | right-easy | right-hard |
|---|---|---|---|---|
| left-easy | – | $< 0.01$ | 0.13 | 0.12 |
| left-hard | | – | 0.14 | 0.13 |
| right-easy | | | – | $< 0.01$ |
| right-hard | | | | – |

## E  Additional Experimental Results

### E.1  Additional Comparison Results

Due to the varying reward ranges across different tasks, the multi-task performance in Table 1 is presented after normalization, which can be formulated as,

$$r_{\mathrm{norm}} = \frac{r_{\mathrm{raw}} - b_{\mathrm{min}}}{b_{\mathrm{max}} - b_{\mathrm{min}}} \times 100. \tag{14}$$

The boundaries in different tasks vary, which are reported in Table 9.

Table 9: The boundaries of tasks in Bi-DexHands.

| Task | $b_{\mathrm{min}}$ | $b_{\mathrm{max}}$ | Task | $b_{\mathrm{min}}$ | $b_{\mathrm{max}}$ | Task | $b_{\mathrm{min}}$ | $b_{\mathrm{max}}$ |
|---|---|---|---|---|---|---|---|---|
| block-stack | 150 | 325 | bottle-cap | 0 | 280 | catch-abreast | 0 | 35 |
| catch-over-underarm | 0 | 25 | catch-underarm | 0 | 15 | close-inward | 0 | 420 |
| close-outward | 0 | 1000 | open-inward | 0 | 400 | open-outward | 0 | 600 |
| kettle | -200 | 50 | lift-underarm | -60 | 275 | over | 0 | 30 |
| pen | 0 | 125 | scissors | -50 | 400 | | | |

We present the comparison results of all tasks in MA-Mujoco in Table 10.

### E.2  Statistical Analysis of Comparison Results

Across the 38 tasks from Bi-DexHands and MA-Mujoco, M3W achieves the best performance on 71% (27 out of 38) of the tasks. Conventionally, achieving the best result on more than 50% of tasks is considered statistically significant [17]. In addition, we conducted statistical analyses of the comparative results shown in Table 1 and Table 10, with detailed outcomes summarized in Table 11. From the perspective of statistical significance, M3W demonstrates significant improvements ($p < 0.05$) on 27 out of 38 tasks. We also employed the Probability of Superiority (PoS) to assess the relative performance of the algorithms. Comparing M3W to the best-performing baselines, M3W shows explicit advantage (PoS > 64%) on 26 tasks, overall advantage (PoS > 50%) on 27 tasks, and explicit disadvantage (PoS < 36%) on only 6 tasks.

Table 10: Comparision results on multi-task MA-Mujoco.

| Suite | Task | M3W(ours) | MACD [1] | HMASD [40] | MAT [38] | MAPPO [41] | HATRPO [19] |
|---|---|---|---|---|---|---|---|
| 2a-Cheetah | run | **31.4 ± 0.5** | 25.6 ± 1.2 | 14.7 ± 3.0 | 5.9 ± 0.3 | 4.3 ± 0.3 | 4.6 ± 0.2 |
| | run-backwards | **33.0 ± 0.9** | 23.5 ± 2.5 | 16.3 ± 3.8 | 6.9 ± 0.3 | 7.8 ± 0.6 | 3.7 ± 0.2 |
| | jump | **40.8 ± 0.9** | 35.6 ± 2.3 | 37.1 ± 1.0 | 18.0 ± 0.6 | 28.7 ± 1.8 | 18.4 ± 0.5 |
| | run-backleg | **32.5 ± 1.0** | 17.7 ± 1.6 | 19.9 ± 2.9 | 5.8 ± 0.7 | 7.2 ± 0.8 | 1.8 ± 0.1 |
| | run-frontleg | **28.5 ± 0.8** | 19.4 ± 2.0 | 20.4 ± 1.4 | 5.9 ± 0.4 | 7.5 ± 0.8 | 3.7 ± 0.3 |
| 6a-Cheetah | run | 9.4 ± 0.4 | 8.6 ± 0.6 | **11.0 ± 1.0** | 7.5 ± 0.7 | 4.6 ± 0.4 | 2.6 ± 0.2 |
| | run-backwards | 12.3 ± 0.4 | 7.5 ± 0.7 | **18.1 ± 0.7** | 14.2 ± 0.5 | 9.8 ± 0.6 | 5.6 ± 0.3 |
| | jump | 39.1 ± 0.3 | 32.8 ± 1.0 | 37.8 ± 0.7 | **41.1 ± 0.7** | 31.9 ± 0.9 | 15.4 ± 0.5 |
| | run-backleg | **16.6 ± 0.3** | 13.0 ± 0.9 | 14.5 ± 0.2 | 13.8 ± 0.4 | 10.9 ± 0.4 | 4.6 ± 0.4 |
| | run-frontleg | 16.6 ± 0.6 | 14.2 ± 0.8 | **18.2 ± 0.3** | 13.5 ± 0.4 | 9.2 ± 0.3 | 3.5 ± 0.3 |
| Humanoid | run | **35.6 ± 1.0** | 9.7 ± 0.7 | 15.6 ± 1.7 | 8.6 ± 0.4 | 12.3 ± 0.4 | 2.9 ± 0.2 |
| | stand | **20.8 ± 0.3** | 10.0 ± 0.5 | 20.8 ± 0.2 | 6.2 ± 0.4 | 10.0 ± 0.2 | 6.9 ± 0.2 |
| | walk | **50.5 ± 2.0** | 10.2 ± 0.6 | 22.7 ± 3.2 | 14.5 ± 0.8 | 20.3 ± 0.6 | 4.8 ± 0.3 |
| | standup | 21.6 ± 0.3 | 20.6 ± 0.6 | 11.1 ± 0.3 | **23.4 ± 0.3** | 12.5 ± 1.8 | 11.0 ± 1.2 |
| Walker2D | run | 22.5 ± 0.4 | 22.6 ± 1.9 | **33.5 ± 4.6** | 5.2 ± 1.2 | 2.8 ± 0.3 | 0.6 ± 0.1 |
| | run-backwards | **24.8 ± 0.4** | 14.3 ± 2.2 | 19.3 ± 1.0 | 7.1 ± 0.5 | 4.1 ± 0.5 | 4.9 ± 0.2 |
| | stand | **90.3 ± 0.8** | 86.0 ± 1.7 | 83.8 ± 1.9 | 60.1 ± 2.5 | 66.4 ± 2.9 | 67.4 ± 3.7 |
| | walk | **71.7 ± 0.4** | 16.7 ± 2.5 | 51.6 ± 2.1 | 26.2 ± 1.9 | 40.1 ± 1.2 | 2.4 ± 0.7 |
| | walk-backwards | **42.1 ± 1.9** | 32.4 ± 1.4 | 26.8 ± 2.6 | 25.8 ± 1.5 | 21.8 ± 1.4 | 23.0 ± 0.9 |
| Swimmer | swim | 46.7 ± 2.6 | 47.2 ± 2.0 | 48.0 ± 1.3 | **49.7 ± 0.7** | 44.5 ± 0.3 | 33.9 ± 5.8 |
| | swim-backwards | 46.6 ± 1.1 | **57.9 ± 1.0** | 51.4 ± 1.5 | 40.4 ± 0.6 | 48.1 ± 0.4 | 11.0 ± 1.4 |
| Hopper | hop | **65.4 ± 0.6** | 37.4 ± 1.6 | 51.4 ± 0.9 | 42.1 ± 1.3 | 35.6 ± 0.8 | 9.1 ± 2.4 |
| | stand | **87.2 ± 1.1** | 77.1 ± 0.9 | 81.0 ± 1.6 | 78.9 ± 0.8 | 76.7 ± 1.0 | 78.3 ± 0.8 |
| Reacher | reach | **87.4 ± 1.0** | 68.2 ± 2.7 | 70.5 ± 1.3 | 70.8 ± 2.6 | 67.7 ± 2.3 | 61.8 ± 2.0 |
| | Average | **40.2 ± 1.1** | 29.5 ± 1.1 | 33.1 ± 1.1 | 24.3 ± 1.0 | 24.1 ± 1.0 | 15.7 ± 1.0 |

Table 11: Statistical analysis across all tasks.

| Task | Baseline | p-value | PoS | Task | Baseline | p-value | PoS |
|---|---|---|---|---|---|---|---|
| **Bi-DexHands** | | | | | | | |
| over | MACD | < 0.001 | 75.2% | bottle-cap | MAPPO | 0.004 | 47.3% |
| catch-abreast | MACD | <0.001 | 77.8% | catch-over2underam | MACD | 0.027 | 72.2% |
| catch-underarm | HMASD | <0.001 | 68.9% | block-stack | MACD | <0.001 | 74.1% |
| door-close-inward | HMASD | <0.001 | 68.1% | door-close-outward | MACD | <0.001 | 72.6% |
| door-open-inward | MACD | 0.069 | 45.5% | door-open-outward | MACD | <0.001 | 89.1% |
| kettle | HMASD | 0.075 | 55.1% | lift-underarm | MACD | < 0.001 | 76.7% |
| pen | MACD | <0.001 | 79.8% | scissors | HMASD | <0.001 | 49.2% |
| **MA-Mujoco** | | | | | | | |
| 2-cheetah-run | MACD | <0.001 | 67.5% | 2-cheetah-runbwd | MACD | <0.001 | 73.5% |
| 2-cheetah-jump | HMASD | <0.001 | 67.9% | 2-cheetah-runback | HMASD | <0.001 | 69.3% |
| 2-cheetah-runfront | HMASD | <0.001 | 79.4% | 6-cheetah-run | HMASD | <0.001 | 36.4% |
| 6-cheetah-runbwd | HMASD | <0.001 | 3.9% | 6-cheetah-jump | MAT | <0.001 | 27.3% |
| 6-cheetah-runback | HMASD | <0.001 | 86% | 6-cheetah-runfront | HMASD | <0.001 | 32.9% |
| humanoid-run | HMASD | 0.032 | 98.9% | humanoid-stand | HMASD | <0.041 | 65.2% |
| humanoid-walk | HMASD | <0.001 | 100.0% | humanoid-standup | MAT | <0.001 | 10.8% |
| walker2d-run | HMASD | <0.001 | 39.3% | walker2d-runbwd | HMASD | <0.001 | 84.9% |
| walker2d-stand | MACD | <0.001 | 75.2% | walker2d-walk | HMASD | <0.001 | 92.8% |
| walker2d-walkbwd | MACD | <0.001 | 100.0% | swimmer-swim | MAT | 0.007 | 31.0% |
| swimmer-swimbwd | MACD | <0.001 | 11.9% | hopper-hop | HMASD | <0.001 | 100.0% |
| hopper-stand | HMASD | <0.001 | 73.4% | reacher-reach | MACD | <0.001 | 90.4% |

**Probability of Superiority (PoS)** is a non-parametric measure used to quantify the likelihood that a randomly selected score from one algorithm (e.g., M3W) is greater than a randomly selected score from another (e.g., a baseline). Formally, given two sets of performance scores $X$ and $Y$, drawn from two algorithms respectively, PoS is defined as:

$$\text{PoS}(X, Y) = P(x > y) + \frac{1}{2}P(x = y), \quad \text{for } x \in X, y \in Y$$

In our analysis, we interpret:PoS > 64% as an *explicit advantage* of one method over another, PoS > 50% as a *overall advantage*, and PoS < 36% as a *explicit disadvantage*. This metric is especially

useful when comparing algorithms across multiple stochastic tasks, as it avoids assumptions about the underlying performance distributions and offers an interpretable measure of practical dominance.

### E.3 Details of Ablation Settings

**Experimental Settings.** We selected six tasks from MA-Mujoco involving two robot suites. In the task set {Cheetah: `run`, `run-bwd`, `run-front`; Walker2D: `run`, `run-bwd`, `walk`}, although the semantics and objectives differ across tasks, their dynamics exhibit bounded similarity. We collected 25K steps of transitions for each task using random actions, resulting in a combined dataset of 150K transitions, which was used consistently across all models for training. Training hyperparameters are the same with Table 3. Models are trained for 2,500 iterations, during which all modules are frozen except the ablated components. Specifically, for dynamics prediction experiments, the unfrozen modules include the encoder, dynamics model, and task embedding, with the loss defined as follows:

$$\mathcal{L}(\theta_E, \theta_D, e) = \sum_i^n \sum_t^H \lambda^t \left\| q_D\left(\boldsymbol{z_t}, \boldsymbol{a_t}, e\right)^i - z_{t+1}^i \right\|_2^2. \tag{15}$$

For reward prediction experiments, the unfrozen modules include the encoder, reward model, and task embedding, with the loss defined as:

$$\mathcal{L}(\theta_E, \theta_R, e) = \sum_t^H \lambda^t \left( q_R\left(\boldsymbol{z_t}, \boldsymbol{a_t}, e\right) - r_{t+1}\right)^2. \tag{16}$$

In Eq. (15) and Eq. (16), the loss is backpropagated through the latent variable, where $z_t^i = p_E(o_t^i, e)$.

**Parameter Sizes.** We report the parameter sizes of all models in Table 12. The results reveal a trade-off between model size and prediction error. M3W increases the parameters count by +12% (dynamics), +76% (reward), but achieves a substantial reduction in prediction error by -53% (dynamics), -78% (reward).

Table 12: Model sizes and prediction errors of different models. ST means Switch Transformer, SA means Self-Attention.

| | | **M3W** | **ST** | **Transformer** | **SA** | **MLP** |
|---|---|---|---|---|---|---|
| Dynamics | Size | 707,072 | 630,768 | 192,016 | - | 35,648 |
| | Error | $0.042 \pm 0.007$ | $0.067 \pm 0.009$ | $0.119 \pm 0.011$ | - | $0.422 \pm 0.058$ |
| Reward | Size | 267,280 | - | - | 151,616 | 33,600 |
| | Error | $0.14 \pm 0.03$ | - | - | $0.59 \pm 0.12$ | $0.81 \pm 0.26$ |

The training curves of M3W across the six tasks are shown in Figure 13. The results reveal that the dynamics error curves naturally cluster into two distinct groups, corresponding to the Cheetah and Walker2D task suites, respectively. This observation further supports our claim regarding the presence of bounded similarity in multi-task dynamics.

### E.4 Additional Ablation Studies on Actor Prior

As shown in Eq.1, M3W incorporates both an actor and a critic, though their importance and roles differ significantly. The actor is used solely to generate candidate actions for the planner and does not act as a policy during decision-making. In contrast, the critic is crucial, as it estimates the expected return of the planning horizon in Eq. (7).

Table 13 presents an ablation study on the actor prior. In the *planner-only* variant, the planner does not rely on the actor prior, and all candidate actions are generated randomly. The results indicate that the actor prior contributes to improving action quality and reducing variance. However, M3W remains effective even without the actor prior, demonstrating the robustness of the planning component.

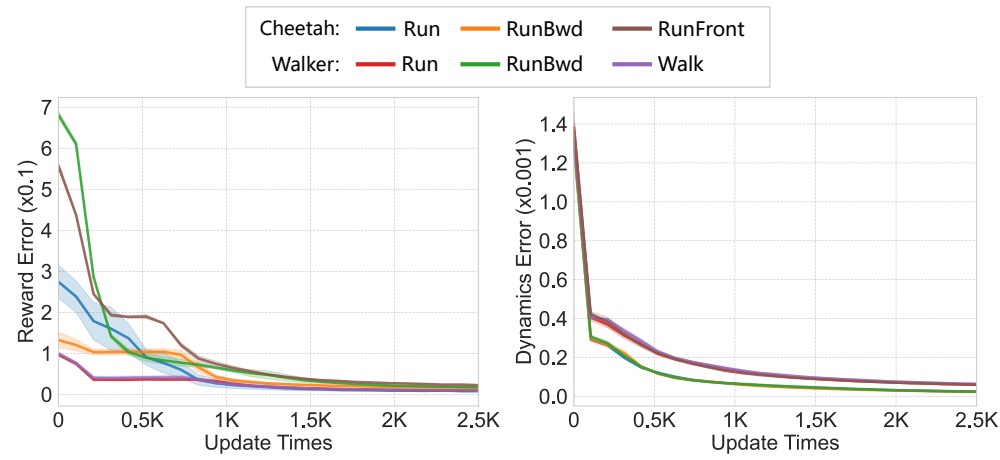

Figure 13: Training curves of M3W in dynamics error and reward error.

Table 13: Ablation studies of actor prior.

| Task | Default | Planner-Only | Task | Default | Planner-Only |
|---|---|---|---|---|---|
| **Bi-DexHands** | | | | | |
| over | 70.0 ± 1.3 | 67.3 ± 1.6 | bottle-cap | 89.8 ± 0.6 | 89.2 ± 0.5 |
| catch-abreast | 64.1 ± 4.4 | 60.3 ± 5.1 | catch-over2underam | 84.8 ± 1.4 | 83.8 ± 2.0 |
| catch-underarm | 87.3 ± 1.3 | 83.6 ± 13.8 | block-stack | 88.1 ± 3.1 | 84.1 ± 2.9 |
| door-close-inward | 91.3 ± 0.7 | 90.2 ± 1.2 | door-close-outward | 95.0 ± 1.2 | 94.8 ± 1.5 |
| door-open-inward | 91.5 ± 0.4 | 91.0 ± 1.6 | door-open-outward | 89.0 ± 2.1 | 87.1 ± 3.2 |
| kettle | 90.4 ± 0.4 | 89.9 ± 0.4 | lift-underarm | 77.0 ± 2.1 | 73.3 ± 2.1 |
| pen | 83.3 ± 3.3 | 76.3 ± 6.8 | scissors | 76.4 ± 2.3 | 69.9 ± 6.6 |
| average | 84.1 ± 0.8 | 81.4 ± 3.5 | | | |
| **MA-Mujoco** | | | | | |
| 2-cheetah-run | 31.4 ± 0.5 | 30.9 ± 0.7 | 2-cheetah-runbwd | 33.0 ± 0.9 | 32.8 ± 0.9 |
| 2-cheetah-jump | 40.8 ± 0.9 | 39.6 ± 1.5 | 2-cheetah-runback | 32.5 ± 1.0 | 31.3 ± 0.8 |
| 2-cheetah-runfront | 28.5 ± 0.8 | 27.9 ± 0.9 | 6-cheetah-run | 9.4 ± 0.4 | 9.2 ± 0.6 |
| 6-cheetah-runbwd | 12.3 ± 0.4 | 12.0 ± 0.8 | 6-cheetah-jump | 39.1 ± 0.3 | 38.5 ± 0.3 |
| 6-cheetah-runback | 16.6 ± 0.4 | 16.3 ± 0.6 | 6-cheetah-runfront | 16.6 ± 0.6 | 16.4 ± 0.6 |
| humanoid-run | 35.6 ± 1.0 | 35.0 ± 1.2 | humanoid-stand | 20.8 ± 0.3 | 20.4 ± 0.3 |
| humanoid-walk | 50.5 ± 2.0 | 50.1 ± 2.6 | humanoid-standup | 21.6 ± 0.3 | 20.9 ± 0.4 |
| walker2d-run | 22.5 ± 0.4 | 22.0 ± 0.6 | walker2d-runbwd | 24.8 ± 0.4 | 24.2 ± 0.4 |
| walker2d-stand | 90.3 ± 0.8 | 88.4 ± 0.9 | walker2d-walk | 71.7 ± 0.4 | 70.7 ± 0.6 |
| walker2d-walkbwd | 42.1 ± 1.9 | 41.2 ± 2.1 | swimmer-swim | 46.7 ± 2.7 | 45.3 ± 3.1 |
| swimmer-swimbwd | 46.6 ± 1.1 | 45.9 ± 0.8 | hopper-hop | 65.4 ± 0.6 | 63.9 ± 1.1 |
| hopper-stand | 87.2 ± 1.1 | 85.9 ± 0.7 | reacher-reach | 87.4 ± 1.0 | 86.0 ± 2.0 |
| average | 40.2 ± 1.1 | 39.4 ± 1.3 | | | |

## E.5 Robustness to Reward Scale

For policy-centric approaches, significant discrepancies in reward scales across tasks can lead to learning imbalance, where the policy is dominated by tasks with larger reward magnitudes. Since M3W includes a reward predictor as an essential component, a natural question arises: *Is M3W robust to variations in reward scale across tasks?*

To assess the robustness of M3W in this regard, we constructed a task set consisting of `hopper-stand` (reward scaled by ×100) and `hopper-hop` (unchanged). The challenge lies in the fact that `stand` is a much simpler behavior but provides a disproportionately high reward, which can easily lead models to fall into a local optimum. The learning curves are shown in Figure 14.

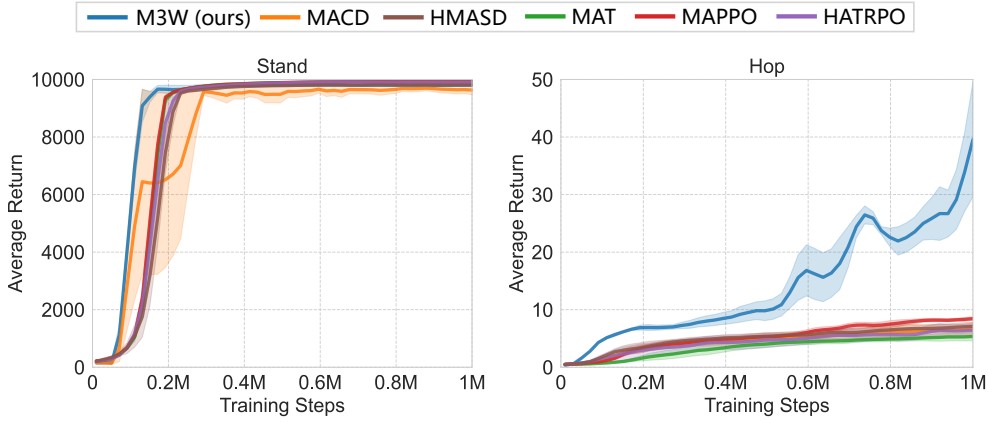

Figure 14: The results validate M3W's robustness to reward scales.

The results show that while all methods converge quickly on the `stand` task, only M3W retains the plasticity to continue learning the more complex `hop` behavior. This demonstrates M3W's robustness to large reward scale discrepancies. Its resilience can be attributed to three key factors:

1. **Reward predictor with task embedding:** M3W's reward predictor takes task embeddings as input, enabling it to differentiate between tasks. As long as each task maintains internally consistent reward scaling, M3W can adjust its predictions accordingly.

2. **Symlog-based numerical transformation:** As described in Appendix C.2, when target values are large in magnitude, the symlog transformation stabilizes training by reducing resolution in high-value ranges.

3. **Planning-based training paradigm:** M3W does not explicitly rely on a policy function, which reduces the risk of becoming trapped in local optima.

### E.6 Time Cost Analysis with an Early-Stopping Planner

**Inference Time Cost.** We report the per-step execution time on a single RTX A6000 GPU (Figure 15, left), where $H$ denotes the rollout horizon and $K_p$ is the number of planner iterations. Under the default settings, M3W achieves an execution time of approximately 25.8 ms per step on the Bi-DexHands environment, which is well below the 50 ms threshold and satisfies the real-time requirements of most robotic tasks.

As shown in Table 14, while policy-centric methods avoid online planning and thus yield lower inference times at the cost of flexibility, our method remains comparable to, or even faster than, existing planner-based approaches [13, 6]. Moreover, the early-stopping heuristic further reduces the inference time to below 10 ms per step, making real-time applications feasible.

Table 14: Per-step inference time (ms) on Bi-DexHands.

| Method | default | early-stop | MACD | HMASD | MAT | MAPPO | HATRPO |
|---|---|---|---|---|---|---|---|
| **Time Cost (ms)** | 25.8 | 9.8 | 1.4 | 3.4 | 2.2 | 1.1 | 1.1 |

**Early-Stopping Planner.** Planning trades additional computation for performance improvement. As the number of iterations increases, action quality and stability typically improve, but so does computational cost. We observe a clear diminishing return: the benefit of additional iterations decreases as action quality improvements taper off. To address this, we propose an early-stopping mechanism based on KL divergence, which monitors the divergence between consecutive action distributions to assess convergence. When the change falls below a predefined threshold, the planner terminates early. As shown on the right side of Figure 15, setting the threshold to 0.5 achieves a 62.1% speedup with only a 1.9% performance drop.

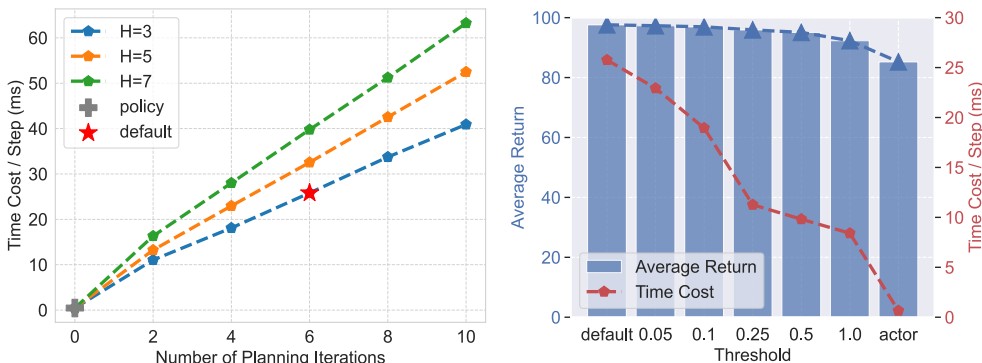

Figure 15: Time cost analysis. (*Left*): Time cost of M3W with different rollout horizons $H$ and planner iterations $K_p$ on Bi-DexHands. (*Right*): Evaluation of average return and time cost for the early-stopping planner.

## E.7 Scalability with the Number of Agents

In M3W, we only concatenate $z_t^i$ and $a_t^i$ for each agent i, not concatenate all agents' spaces together. So it prevents a dimensional explosion as the number of agents increases. To validate the scalability of M3W, we conducted evaluations on Many-Agent-Mujoco and MAgent [48], with results shown in Figure 16.

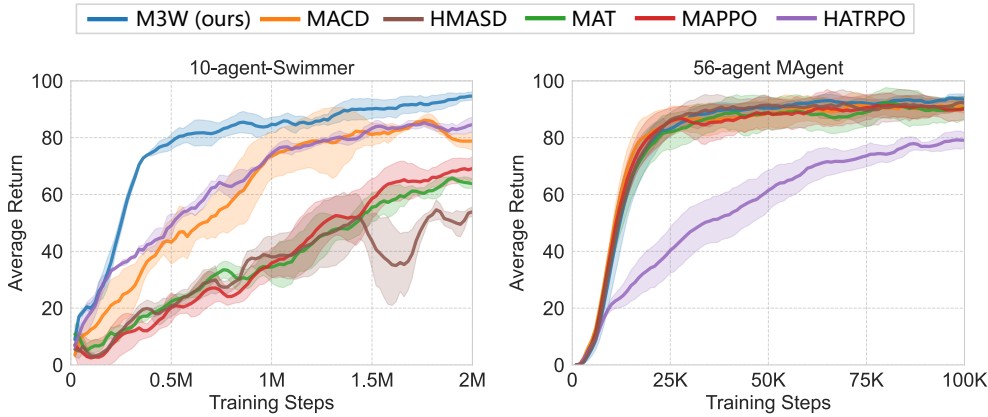

Figure 16: Additional Comparisons on 10-agent-Swimmer and 56-agent MAgent, which confirm the scalability of M3W with the number of agents.

