# OpenReview forum: "Learning and Planning Multi-Agent Tasks via an MoE-based World Model"
_NeurIPS.cc/2025/Conference — NeurIPS 2025 poster_

### Official Review · Reviewer_k53v · 2025-06-29

**Clarity:** 2
**Significance:** 3
**Originality:** 3
**Rating:** 5
**Confidence:** 3

**Summary:**

The paper presents M3W, a novel approach for multi-task multi-agent reinforcement learning that applies a mixture-of-experts (MoE) structure to the world model instead of the policy. It identifies the phenomenon of bounded similarity in multi-task dynamics and leverages this property to improve learning efficiency and planning performance.

**Questions:**

1. Could you provide more detailed explanation on why the actor in the world model is not used as an explicit decision-making policy? What are the advantages and disadvantages of using it solely for generating candidate actions for the planner compared to using it as a direct policy? Is there any experimental comparison?
2. How does the training cost and inference time of the world model compare to policy-based methods? Could you provide some analysis or experimental results on this aspect?

**Ethical Concerns:**

["NO or VERY MINOR ethics concerns only"]

**Final Justification:**

I have increased my score, as the authors have addressed my concerns.

**Limitations:**

The authors have mentioned some limitations of the current work, such as the applicability to continuous action spaces only. However, more discussion on the potential limitations regarding the scalability of the world model would be beneficial.

**Paper Formatting Concerns:**

No major formatting issues found.

**Quality:**

3

**Strengths And Weaknesses:**

* Strengths:
1. Proposes a new perspective on modularity in multi-task learning by applying MoE to the world model, which effectively captures bounded similarity among tasks.
2. Demonstrates superior performance and sample efficiency in experiments on Bi-DexHands and MA-Mujoco.
3. Provides comprehensive ablation studies and visualizations to support the effectiveness of the MoE-based architecture.
* Weaknesses:
1. Limited discussion on the training cost and inference time of the world model compared to policy-based methods, which is crucial for understanding the practicality of the approach.
2. The generalizability of the world model to other scenarios is not thoroughly analyzed.

---

> ### Author Rebuttal · Authors · 2025-07-27
>
> `Responses`
> ---
> ---
>
> > **Q1: The training cost and inference time of the world model compare to policy-based methods. (Weakness.1, Question.2)**
>
> **A1:** As reported in (E.6 *Time Cost Analysis with an Early-Stopping Planner*, L649, Appendix), we have already provided the inference time of our method. Following your suggestions, we additionally report the training and inference time of the baselines in the table below:
>
> ||Ours(default)|Ours(early-stop)|MACD|HMASD|MAT|MAPPO|HATRPO
> |-|-|-|-|-|-|-|-
> Inference (ms)|25.8|9.8|1.4|3.4|2.2|1.1|1.1
> Train (h)|~19||~34|~14|~11|~8|~14
>
> Note that planning-based methods inherently trade additional computation for improved performance, which aligns with our design objectives. Compared to policy-centric baselines, our method incurs higher computational cost but achieves substantially better sample efficiency and performance. Furthermore, its inference time is comparable to, or even faster than, other existing planner-based approaches [1,2]. Moreover, our early-stopping heuristic further reduces the inference time to **< 10 ms/step**, making real-time applications feasible.
>
> &nbsp;
>
> ---
>
> > **Q2: The generalizability of the world model to other scenarios is not thoroughly analyzed. (Weakness.2)**
>
> **A2:** The factors that hinder the generalization of our method to other scenarios include the full observation assumption and the continuous action space limitation.
>
> + **Partial Observability:** We acknowledge that the full-observability assumption is indeed a common limitation of existing model-based MARL methods [3,4]. However, our method can be naturally extended to partial observability by two simple modifications:
>
>   1. **Training-time masking:** Inspired by Masked AutoEncoder [5] (and its applications in MARL [6]), we randomly mask other agents with a certain probability to simulate communication failures, thereby enhancing robustness.
>
>   2. **Planning-time communication cache:** The planner caches previously predicted rollouts and reuses them in case of communication failure, replacing missing ground-truth inputs to maintain trajectory continuity.
>
>   + **Preliminary results:** We tested the above extensions on *BiDexHands-Over* and confirmed their effectiveness. In *mask* group, other agents are masked with 50% probability, and communication fails with 50% probability during execution. **These results show that our method can adapt to partial observability with minimal performance degradation.**
>
>     ||full|mask|decentralized|
>     -|-|-|-
>     Dynamics Error ↓|0.025±0.003|0.033±0.005|0.044±0.005
>     Reward Error ↓|0.004±0.001|0.012±0.001|0.013±0.001
>     Return ↑|32.1±0.4|31.3±0.6|31.1±0.3
>
> + **Continuous Action Space:** There are two modifications to extend our method to discrete action spaces:
>   1. Use one-hot actions or bitactions [6] to transform a discrete action into a continuous one.
>   2. Replace the MPPI control with Cross Entropy Method (CEM) or Monte Carlo Tree Search (MCTS).
>
>   During training, discrete actions are converted into one-hot representations to facilitate understanding by the world model. During execution, a CEM-based planner is adopted, and its procedure is reported below. Similar to the above, we evaluated these extensions on the classic discrete tasks SMAC ‘3m’ and RWARE ‘left-easy’, achieving a **100% win rate** and a reward **> 7.75/100 steps**.
>
>   **CEM Procedure:** Assume the action dimension is $n$,
>   + S1: Initialize a discrete action distribution, $p^{(0)}=1/n$
>   + S2: Sample $M$ action sequences, and compute their values according to (Eq.7, L185, main text)
>   + S3: Select the top $m$ elite action sequences $a^∗$
>   + S4: For each action dimension $j$, update the distribution $p^{(k+1)}(a[j])=\frac{1}{m} \sum \mathbb{I}(a[j]=a^*)$
>   + Repeat steps S2–S4 until convergence.
>
> &nbsp;
>
> ---
>
> > **Q3: Why the actor is not used as an explicit policy? What are the advantages and disadvantages? (Question.1)**
>
> **A3:**
>
> 1. **Reason:** We believe **the dynamics model to be more reliable and transferable across tasks than the actor.** As you noted in your Summary and Strengths.1, the bounded similarity phenomenon implies that different tasks share similar underlying dynamics but have highly diverse policies, making the learned actor less robust for multi-task generalization.
>
> 2. **Advantages:** The planner achieves considerably higher performance. (E.4, *Additional Ablation Studies on Actor Prior*, L620, Appendix) compared planner performance when using actor-generated candidate actions versus random candidates. Following your suggestions, we additionally included a direct actor-only policy as a baseline. Results are shown below:
>
>     ||actor+plan|plan-only|actor-only
>     -|-|-|-
>     Return ↑| **84.1 ± 0.8** | 81.4 ± 3.5 | 76.2 ± 2.1
>
> 3. **Disadvantages:** Directly using the actor as the decision-making policy offers better time efficiency. To mitigate the planner’s higher time cost, we introduced an early-stopping heuristic in (E.6 *Early-Stopping Planner*, L655, Appendix), which terminates planning when the KL-divergence between consecutive action distributions falls below a threshold. With a threshold of 0.5, **it achieved a 62.1% speedup with only a 1.9% performance drop.**
>
> These results show that the actor serves best as an action generator for the planner, while the planner remains the primary decision-making module for robust multi-task performance.
>
> &nbsp;
>
>
> `Summary`
> ---
> ---
>
> Thank you for your valuable review and constructive feedback. We appreciate your recognition of the **motivation**, **technical soundness** and **strong performance** of our work.
>
> We also acknowledge the affirmations from other reviewers R1(DXnW), R2 (gkLA), and R3 (mmFi) on:
>
> + **Motivation and novelty:** (**R1:** *moves modularity from policy to world model*, *first introduce MoE-based world model for multi-task setting*), (**R2:** *identifies the novel insight of ‘bounded similarity’*; *introduces an MoE-based world model … a conceptual shift in MT-MARL*), (**R3:** *well-structured and strongly motivated*).
>
> + **Method design:** (**R1:** *task-conditioned routing reduces gradient interference*, *load-balance loss enhances training robustness*), (**R2:** *allows better exploitation of inter-task structure;* *modular components selectively share or isolate knowledge*).
>
> + **Strong performance:** (**R1:** *achieves the top return*, *with significant improvements over baselines*), (**R2:** *outperforms baselines in both return and sample efficiency*), (**R3:** *really strong results*).
>
> + **Interpretability:** (**R1:** *enhancing interpretability through expert specialization*, *visualizations strengthen the interpretability of M3W*), (**R3:** *interpretability analysis shows the mechanism of sharing experts*).
>
> We hope that our responses have fully addressed your concerns, and we welcome any further suggestions to improve this work. **If our clarifications have resolved your concerns, we would sincerely appreciate your consideration for raising the score.**
>
>
> &nbsp;
>
> `References`
> ---
> ---
>
> [1] TD-MPC2: Scalable, Robust World Models for Continuous Control, ICLR, 2024.
>
> [2] Sparse Imagination for Efficient Visual World Model Planning, arXiv, 2025.
>
> [3] Aligning Credit for Multi-Agent Cooperation via Model-based Counterfactual Imagination, AAMAS, 2024.
>
> [4] Decentralized Transformers with Centralized Aggregation are Sample-Efficient Multi-Agent World Models, TMLR, 2025.
>
> [5] Masked Autoencoders Are Scalable Vision Learners, CVPR, 2022.
>
> [6] MA2E: Addressing Partial Observability in Multi-Agent Reinforcement Learning with Masked Auto-Encoder, ICLR, 2025.
>
> [7] INS: Interaction-aware Synthesis to Enhance Offline Multi-agent Reinforcement Learning, ICLR, 2025.

---

> ### Comment · Reviewer_k53v · 2025-08-01
>
> I appreciate the authors' effort in the rebuttal and they addressed my concerns and confusion. I believe if contents of rebuttal could be incorporated into the the paper, it could really help paint a better picture of the contribution and findings.  Consequently, I will raise my rating.

---

> ### Author Response · Authors · 2025-08-01
> **Thank You and a Quick Check on Rating Update**
>
> Dear Reviewer k53v,
>
> Thank you very much for your kind follow-up and for acknowledging our rebuttal. We truly appreciate your recognition of our efforts and your willingness to raise the rating. We would also like to reaffirm that all the clarifications and improvements in the rebuttal will be carefully incorporated into the camera-ready version.
>
> **As a quick note, we noticed that the review interface currently does not reflect an updated score ( it is still shown as 4 ). If convenient, could you kindly double-check whether the rating has been successfully submitted on your end (via the "Edit" button in the Official Review section)?**
>
> We sincerely appreciate your time and support throughout the review process.
>
> Best regards,
>
> Authors of Submission#6412

---

### Official Review · Reviewer_mmFi · 2025-07-01

**Clarity:** 3
**Significance:** 3
**Originality:** 2
**Rating:** 4
**Confidence:** 3

**Summary:**

This paper discusses the paradigm of Multi-Task Multi-Agent Reinforcement Learning (MT-MARL) where a group of agents are cooperatively solving a set of tasks while employing a single model. In this paper, the authors focused on modeling a world model instead of a centralized policy by utilizing a mixture of experts architecture that helps in sharing knowledge between similar tasks while disentangling sharing across dissimilar ones. This work is motivated by the phenomenon, referred to as bounded similarity, where a group of similar tasks exhibits similarity in their underlying dynamics, while across groups dissimilarities appear. The proposed approach, named M3W, has been evaluated on Bi-DexHands and MA-Mujoco, showing strong performance and sample efficiency. In addition, interpretability analysis has been conducted to show the mechanism of sharing experts among a certain group of similar tasks.

**Questions:**

- What is the effect of scaling the model down or up (in terms of the number of experts) with the number of tasks or the number of agents?

**Ethical Concerns:**

["NO or VERY MINOR ethics concerns only"]

**Final Justification:**

Since the authors addressed my concerns and questions, I will increase my score.

**Limitations:**

- I believe a major limitation of the proposed method, which I believe this work did not discuss well, is the scalability of this approach due to the high parameter count. I understand the improved results by this method, but this raises the question of whether this is due to the capacity of the model and not the design choices.

**Paper Formatting Concerns:**

No major formatting issues in this paper

**Quality:**

3

**Strengths And Weaknesses:**

# Strengths

- In my opinion, this paper is well-structured and strongly motivated. In addition, I appreciate the flow of the paper and the quality of writing.
- The proposed method connects recent advances in supervised learning (e.g, using SoftMoE and SparseMoE) and exploits their benefits in an interesting problem in RL.
- The proposed method sounds and demonstrates really strong results on two different benchmarks.

# Weaknesses

- The proposed method utilizes a complex structure with many parameters. In the appendix, the parameter count is listed for each method, showing the computational burden of the proposed method.
- The demonstrated concept and tools are known and not original. Yet, the technical aspect of the paper and the presented results dominate that.
- One important ablation study is missing, which might have a strong impact on the work. An analysis of the number of experts needed is essential to understand the scalability of this method. I believe that not discussing it is affecting the fairness of the evaluation.
- Another important weakness to highlight is the weak and insufficient discussion of the MTRL literature and the MoE in the MTRL literature. Since the problem tackled in this work is MT-MARL, the literature of each component should have been discussed in more depth. For instance, many works [1, 2, 3, 4] that utilize MoE or multiple models in MTRL were never mentioned.
- Minor typos and errors, in Line 196, $q_D$ should be predicting the next latent state $z_{t+1}^i$ instead of the current. In Line 205, double "on".

[1] Sodhani, Shagun, Amy Zhang, and Joelle Pineau. "Multi-task reinforcement learning with context-based representations." International Conference on Machine Learning. PMLR, 2021.

[2] Sun, Lingfeng, et al. "Paco: Parameter-compositional multi-task reinforcement learning." Advances in Neural Information Processing Systems 35 (2022): 21495-21507.

[3] Cheng, Guangran, et al. "Multi-task reinforcement learning with attention-based mixture of experts." IEEE Robotics and Automation Letters 8.6 (2023): 3812-3819.

[4] Hendawy, Ahmed, Jan Peters, and Carlo D'Eramo. "Multi-Task Reinforcement Learning with Mixture of Orthogonal Experts." The Twelfth International Conference on Learning Representations.

---

> ### Author Rebuttal · Authors · 2025-07-27
>
> `Responses`
> ---
> ---
>
> > **Q1: Whether the improvements are due to the capacity of the model rather than the design choices? (Weakness.1, Limitation)**
>
> **A1:** We agree that increased parameter count contributes to some extent, but **the main driver is the modular design**. To further disentangle the effect of modularity from parameter scaling, we conducted an experiment similar to Figure 5 in [1]. We keep the number of experts $m$ fixed but reduce the dimension of each expert to $\frac{1}{m}$.
>
> Following the experimental settings in (E.3 *Details of Ablation Settings*, L601, Appendix), the results shows that, SoftMoE achieves significantly lower prediction error compared to a dense MLP model with a similar parameter count. **These results indicate that performance gains primarily stem from the modular design, not merely from increasing parameters.**
>
> |Model|Parameters|Prediction Error
> -|-|-
> SoftMoE|707,072|0.042±0.007
> SoftMoE(÷m)|127,232|0.064±0.016
> Densen|128,290|0.213±0.051
>
> &nbsp;
>
> ---
>
> > **Q2: Missing an analysis of the number of experts. (Weakness.3, Question)**
>
> **A2:** Following your constructive suggestion, we conducted an additional ablation study on the number of experts, and the results show:
>
> + **Dynamics prediction:** The dynamics error decreases rapidly as the number of experts increases but plateaus after ~16 experts across all task scales. This is because SoftMoE’s fully differentiable routing effectively utilizes additional experts while maintaining stable performance.
>
> + **Reward prediction:** The reward error first decreases and then increases as the number of experts grows. Too few experts underfit the reward function, whereas too many slow down training due to sparse routing.
>
> Detailed numerical results are provided in the following table.
>
> | | 6 tasks | | 12 tasks| | 24 tasks| |
> |-|-|-|-|-|-|-|
> |Experts| Dynamics | Reward | Dynamics | Reward | Dynamics | Reward |
> | 4 | 0.043±0.007 | 0.218±0.070 | 0.061±0.010 | 0.421±0.131 | 0.113±0.019 | 0.644±0.198 |
> | 8 | 0.043±0.007 | **0.114±0.027** | 0.048±0.008 | 0.211±0.084 | 0.098±0.017 | 0.502±0.113 |
> | 16| **0.042±0.007** | 0.147±0.035 | **0.046±0.007** | **0.151±0.039** | 0.067±0.011 | **0.185±0.054** |
> | 24| 0.043±0.007 | 0.176±0.043 | 0.048±0.007 | 0.217±0.086 | 0.062±0.010 | 0.268±0.094 |
> | 32| **0.042±0.007** | 0.209±0.049 | **0.046±0.007** | 0.288±0.093 | **0.060±0.010** | 0.365±0.101 |
>
> &nbsp;
>
> ---
>
> > **Q3: The demonstrated concept and tools are known and not original. (Weakness.2)**
>
> **A3:** Thank you for recognizing the technical soundness of our work. We would like to explain that the main novelty of our paper lies in **its conceptual and methodological contributions** rather than introducing new tools or network architectures. Specifically:
>
> + **Bounded similarity:** We identify the phenomenon of bounded similarity in multi-task settings, offering a new perspective on modeling task relationships — as emphasized by **R1** (*the authors point out … called bounded similarity*) and **R2** (*identifies the novel insight of ‘bounded similarity’*).
>
> + **MoE-based world model:** We apply MoE to the world model instead of the policy, introducing a new paradigm for modularity in MT-MARL — as emphasized by **R2** (*conceptual shift in MT-MARL*) and **R4** (*a new perspective on modularity in MTRL*).
>
> &nbsp;
>
> ---
>
> > **Q4: Insufficient discussion of MTRL and MoE-related literature, plus minor typos. (Weakness.4, Weakness.5)**
>
> **A4:** Thank you for your careful and constructive review. Following your suggestion, we will expand the related work section to include recent MoE-based MTRL approaches and clarify their differences from our work.
>
> We also appreciate your attention. The typos in Line 196 and Line 205 have been corrected, and we will carefully proofread the entire paper to avoid similar errors.
>
> The following are the relevant paragraphs that are expected to be added：
>
> > As model capacity grows and generalization becomes more important, Mixture of Experts (MoE) has gained attention in multi-task reinforcement learning (MTRL) for balancing shared and task-specific knowledge.
> >
> > CARE[2] proposed context-based representations to differentiate tasks within a shared policy, laying the foundation for the integration of MoE in MTRL. Building on this, Paco[3] introduced a parameter-compositional MTRL method, which  combines shared and task-specific parameters, enhancing knowledge transfer. More structurally aligned with traditional MoE, AMESAC[4] proposed an attention-based expert selection mechanism, dynamically selecting different experts to handle task heterogeneity. Furthermore, MOORE[5] developed a MoE framework with orthogonal constraints, promoting functional decoupling among experts to improve generalization and sample efficiency.
>
>
> &nbsp;
>
> `Summary`
> ---
> ---
>
> Thank you for your valuable review and constructive feedback. We appreciate your recognition of the **motivation**, **technical soundness** and **strong performance** of our work.
>
> We also acknowledge the affirmations from other reviewers R1(DXnW), R2 (gkLA), and R4 (k53v) on:
>
> + **Motivation and novelty:** (**R1:** *moves modularity from policy to world model*, *first introduce MoE-based world model for multi-task setting*), (**R2:** *identifies the novel insight of ‘bounded similarity’*; *introduces an MoE-based world model … a conceptual shift in MT-MARL*), (**R4:** *proposes a new perspective on modularity*).
>
> + **Method design:** (**R1:** *task-conditioned routing reduces gradient interference*, *load-balance loss enhances training robustness*), (**R2:** *allows better exploitation of inter-task structure;* *modular components selectively share or isolate knowledge*), (**R4:** *MoE world model effectively captures bounded similarity among tasks*).
>
> + **Strong performance:** (**R1:** *achieves the top return*, *with significant improvements over baselines*), (**R2:** *outperforms baselines in both return and sample efficiency*), (**R4:** *comprehensive ablation studies and visualizations*; *superior performance and sample efficiency*).
>
> + **Interpretability:** (**R1:** *enhancing interpretability through expert specialization*, *visualizations strengthen the interpretability of M3W*).
>
> We hope that our responses have fully addressed your concerns, and we welcome any further suggestions to improve this work. **If our clarifications have resolved your concerns, we would sincerely appreciate your consideration for raising the score.**
>
> &nbsp;
>
> `References`
> ---
> ---
>
> [1] Mixtures of Experts Unlock Parameter Scaling for Deep RL, ICML, 2024.
>
> [2] Multi-task reinforcement learning with context-based representations, ICML, 2021.
>
> [3] Paco: Parameter-compositional multi-task reinforcement learning, NeurIPS, 2022.
>
> [4] Multi-task reinforcement learning with attention-based mixture of experts, IEEE RAL, 2023.
>
> [5] Multi-Task Reinforcement Learning with Mixture of Orthogonal Experts, ICLR. 2024.

---

> > ### Comment · Reviewer_mmFi · 2025-08-04
> >
> > I would like to thank the authors for addressing my questions and concerns.
> >
> > Thank you as well for providing the experiment on reducing the number of parameters in Q1 (and Q2 as well). However, I was wondering why the results report the prediction error instead of the performance in the environments, as shown in Table 1. It is difficult to interpret how the prediction error translates to actual performance. I would appreciate it if the authors could include performance metrics to better evaluate the impact.

---

> ### Author Response · Authors · 2025-08-04
>
> Thank you for your thoughtful follow-up and your valuable suggestions.
>
> Regarding your question about why we reported prediction error instead of environment performance: the primary reason is that evaluating performance (i.e., return) requires running a complete training loop involving planning-based rollouts in the environment, which is significantly more time-consuming than computing prediction errors. Given the time constraints of the rebuttal phase, we chose prediction error as a more efficient proxy for model quality.
>
> Importantly, as our method relies on model-based planning, the accuracy of dynamics and reward prediction directly impacts downstream performance. **A more accurate world model enables more reliable rollouts, which in turn leads to better planning outcomes.**
>
> That said, we fully agree that including performance metrics would make the analysis more comprehensive. Therefore, we report the corresponding average returns in the table below.
>
> We thank the reviewer for the constructive feedback and promise that, if the paper is accepted, we will include these additional experiments in the camera-ready version.
>
>
> |Model|Returns
> |-|-|
> SoftMoE| **35.3 ± 0.9** |
> SoftMoE(÷m)| 33.4 ± 0.8 |
> Densen| 19.9 ± 1.3|
>
> | |6 tasks | 12 tasks| 24 tasks|
> |-|-|-|-|
> | 4 | 33.8 ± 0.8     | 35.7 ± 0.9     | 29.6 ± 0.7     |
> | 8 | **35.9 ± 0.9** | 41.9 ± 1.1     | 34.7 ± 0.9     |
> | 16| 35.3 ± 0.9     | **45.1 ± 1.2** | 40.2 ± 1.1     |
> | 24| 31.9 ± 0.8     | 43.8 ± 1.1     | **41.6 ± 1.1** |
> | 32| 32.3 ± 0.8     | 44.6 ± 1.1     | 41.1 ± 1.2     |

---

> ### Author Response · Authors · 2025-08-08
>
> We would like to thank you again for your constructive feedback.
>
> **Following our previous commitment, we have now completed the additional experiments to evaluate the performance metrics (average return) corresponding to the number of experts.** The new results, which have been added to our comment, confirm that the trends observed in prediction error indeed correlate with performance.
>
> We hope this more complete analysis, which was prompted by your excellent feedback, addresses your concerns and provides a more thorough understanding of our work, and we welcome any further suggestions and disscussions. **With this in mind, we would appreciate it if you would consider raising the score.**

---

### Official Review · Reviewer_gkLA · 2025-07-03

**Clarity:** 3
**Significance:** 3
**Originality:** 3
**Rating:** 5
**Confidence:** 3

**Summary:**

The paper proposes M3W, a novel framework for multi-task multi-agent reinforcement learning (MT-MARL) that leverages a Mixture-of-Experts architecture applied to the world model instead of the policy. M3W is motivated by the observation of bounded similarity across task dynamics—tasks can share highly similar dynamics within groups but differ greatly across groups. To exploit this, M3W employs a SoftMoE-based dynamics model and a SparseMoE-based reward predictor, allowing for modular learning that promotes knowledge reuse across similar tasks while isolating dissimilar ones to prevent gradient interference. Unlike policy-centric approaches, M3W performs model-based planning by generating imagined trajectories from the world model and directly optimizing actions without an explicit policy.

**Questions:**

1/ The model assumes that each agent can access the observations and actions of all other agents through communication. However, in real-world decentralized or partially observable environments (e.g., autonomous driving), such global observability is often infeasible due to sensing limitations and communication constraints. How would M3W perform in more realistic settings with limited or no inter-agent communication? Could the authors discuss possible extensions to support decentralized training or partial observability?

2/  There appears to be a mismatch between the loss formulation in Eq. (2), which uses mean squared error for reward and Q-value regression, and the actual implementation described in Appendix C.1, which uses discretized two-hot encoding and soft cross-entropy loss. Could the authors clarify this discrepancy? Would it be possible to revise Eq. (2) to reflect the actual training objective for better transparency and reproducibility?

**Ethical Concerns:**

["NO or VERY MINOR ethics concerns only"]

**Final Justification:**

The authors have thoroughly addressed my questions.

**Limitations:**

Yes. The author(s) identified the limitation that their method is only applicable to continuous action spaces.

There may be additional limitations, as listed in the "Strengths and Weaknesses" section, that the author(s) could consider including in the paper.

**Paper Formatting Concerns:**

I have no concerns regarding the paper formatting.

**Quality:**

3

**Strengths And Weaknesses:**

Strengths:

1/ The paper introduces a MoE architecture within the world model instead of applying modularity to policies (as most prior works do). This is (to my best knowledge) a conceptual shift in multi-task multi-agent reinforcement learning. This enables both dynamics prediction and reward modeling to be modular and task-adaptive, allowing better exploitation of inter-task structure.

2/ The paper identifies the novel insight of 'bounded similarity', where task dynamics are locally similar within certain groups but globally different across others, and effectively uses this to design modular components that selectively share or isolate knowledge.

3/ Regarding experiments: The method achieves state-of-the-art performance on two challenging benchmarks: Bi-DexHands and MA-Mujoco. It outperforms policy-centric baselines like MAPPO, MAT, and HMASD in both average return and sample efficiency.

Weaknesses:

These are potential areas for future improvement and do not necessarily represent weaknesses:

1/ The model assumes that each agent can access the observations and actions of all other agents through communication (e.g., a shared communication round). However, this assumption may not hold in real-world decentralized or partially observable environments. For example, in autonomous driving scenarios with multiple vehicles (agents), each vehicle typically has limited local sensing and cannot directly observe the internal states or actions of nearby vehicles due to communication delays, privacy constraints, or sensor occlusions. This makes full observability and synchronized communication impractical, limiting the direct applicability of M3W to such domains.

2/ A potential inconsistency arises between the loss formulation Eq. (2) and the actual implementation details described in Appendix C.1. Eq. (2) expresses the reward and Q--alue losses as standard mean squared errors, implying scalar regression, the appendix clarifies that both targets are discretized using a two-hot encoding scheme and trained via soft cross-entropy loss. This discrepancy suggests that the formulation in Eq. (2) does not accurately reflect the implementation.

To improve transparency, the authors should revise or annotate Eq. (2) to indicate the use of discretized targets and soft classification loss.

---

> ### Author Rebuttal · Authors · 2025-07-27
>
> `Responses`
> ---
> > **Q1: How can M3W adapt to decentralized or partially observable settings given its reliance on full observability? (Weakness.1 & Question.1)**
>
> **A1:** We acknowledge that the full-observability assumption is indeed a common limitation of existing model-based MARL methods [1,2]. However, M3W can be naturally extended by two simple modifications:
>
> 1. **Training-time masking:** Inspired by Masked AutoEncoder [3] (and its applications in MARL [4]), we randomly mask other agents with a certain probability to simulate communication failures, thereby enhancing robustness.
>
> 2. **Planning-time communication cache:** The planner caches previously predicted rollouts and reuses them in case of communication failure, replacing missing ground-truth inputs to maintain trajectory continuity.
>
> We tested the above extensions on *BiDexHands-Over* and confirmed their effectiveness. In *mask* group, other agents are masked with 50% probability, and communication fails with 50% probability during execution. **These results show that M3W can adapt to partial observability and limited communication with minimal performance degradation.**
>
> ||full|mask|decentralized|
> -|-|-|-
> Dynamics Error ↓|0.025±0.003|0.033±0.005|0.044±0.005
> Reward Error ↓|0.004±0.001|0.012±0.001|0.013±0.001
> Return ↑|32.1±0.4|31.3±0.6|31.1±0.3
>
> Here are some formulation details of the communication cache.
>
> At any time step $t$, the planner receives the observation $o^i\_t$ and $o^{-i}\_t$, which are encoded into ground-truth latent states $z^i\_t$ and $z^{-i}\_t$ ($-i$ denotes other agents). Starting from $z^{i}\_t$ and $z^{-i}\_t$, the world model performs a rollout to generate a latent trajectory of length $H$, $\Gamma\_1 = \\{\hat{z}^i\_{t+1:t+H}, \hat{z}^{-i}\_{t+1:t+H}\\}$. Ideally, at time $t+1$, the planner would again retrieve $o^i\_{t+1}$, $o^{-i}\_{t+1}$ from the environment and generate a new rollout, $\Gamma\_2 = \\{\hat{z}^i\_{t+2:t+H+1}, \hat{z}^{-i}\_{t+2:t+H+1}\\}$.
>
> Note that $\Gamma\_1$ and $\Gamma\_2$ share overlapping predictions. Therefore, we store $\Gamma\_1$ in a cache, and when communication fails at $t+1$, the planner reuses the cached prediction $\hat{z}^{-i}\_{t+1}$ to replace the missing ground-truth $z^{-i}\_t$. Therefore, the model has the ability to cope with communication interruptions.
>
> &nbsp;
>
> ---
> > **Q2: Discrepancy between Eq.(2)’s MSE loss and the implementation’s two-hot cross-entropy. (Weakness.2 & Question.2)**
>
> **A2:** As you pointed out, the description in (C.1 *Discrete Regression of Reward and Value*, L493, Appendix) accurately reflects the actual implementation. We will update Eq.(2) in the revised version as follows:
>
> $$\mathcal{L}(\theta, \psi) = \sum\_i^n \sum\_t^H \lambda^t \left(
> \underbrace{\left\| \hat{z}^i\_{t+1} - z^i\_{t+1} \right\|\_2^2}\_{\textrm{Dynamic Loss}} + \underbrace{\textrm{SoftCE}\left( \hat{r}\_{t+1} - r\_{t+1} \right)}\_{\textrm{Reward Loss}} + \underbrace{\textrm{SoftCE}\left( \hat{q}\_t - G\_t \right)}\_{\textrm{Q Loss}}
> \right)$$
>
> where,  $\hat{z}^i\_{t+1}=q\_D(\mathbf{z\_t}, \mathbf{a\_t}, e)^i, \hat{r}^i\_{t+1}=q\_R(\mathbf{z\_t}, \mathbf{a\_t}, e), \hat{q}\_t = Q(\mathbf{z\_t}, \mathbf{a\_t}, e)$
>
> &nbsp;
>
>
> `Summary`
> ---
>
> Thank you for your valuable review and constructive feedback. We appreciate your recognition of the **motivation**, **model design** and **strong performance** of our work.
>
> We also acknowledge the affirmations from other reviewers R1(DXnW), R3 (mmFi), and R4 (k53v) on:
>
> + **Motivation and novelty:** (**R1:** *moves modularity from policy to world model*, *first introduce MoE-based world model for multi-task setting*), (**R3:** *well-structured and strongly motivated*), (**R4:** *proposes a new perspective on modularity*).
>
> + **Method design:** (**R1:** *task-conditioned routing reduces gradient interference*, *load-balance loss enhances training robustness*), (**R4:** *MoE world model effectively captures bounded similarity among tasks*).
>
> + **Strong performance:** (**R1:** *achieves the top return*, *with significant improvements over baselines*), (**R3:** *really strong results*), (**R4:** *comprehensive ablation studies and visualizations*; *superior performance and sample efficiency*).
>
> + **Interpretability:** (**R1:** *enhancing interpretability through expert specialization*, *visualizations strengthen the interpretability of M3W*), (**R3:** *interpretability analysis shows the mechanism of sharing experts*).
>
> We hope that our responses have fully addressed your concerns, and we welcome any further suggestions to improve this work. **If our clarifications have resolved your concerns, we would sincerely appreciate your consideration for raising the score.**
>
> &nbsp;
>
> `References`
> ---
> ---
>
> [1] Aligning Credit for Multi-Agent Cooperation via Model-based Counterfactual Imagination, AAMAS, 2024.
>
> [2] Decentralized Transformers with Centralized Aggregation are Sample-Efficient Multi-Agent World Models, TMLR, 2025.
>
> [3] Masked Autoencoders Are Scalable Vision Learners, CVPR, 2022.
>
> [4] MA2E: Addressing Partial Observability in Multi-Agent Reinforcement Learning with Masked Auto-Encoder, ICLR, 2025.

---

> > ### Comment · Reviewer_gkLA · 2025-08-05
> >
> > The authors have thoroughly addressed my questions, and I have updated the score accordingly.

---

### Official Review · Reviewer_DXnW · 2025-07-06

**Clarity:** 3
**Significance:** 3
**Originality:** 3
**Rating:** 5
**Confidence:** 3

**Summary:**

This paper presents M3W, a mixture-of-experts world model for multi-task, multi-agent reinforcement learning. The authors point out that different tasks often need very different policies yet still share parts of the same physics, a phenomenon they call bounded similarity. To use this overlap, they place modularity in the world model instead of in the policy. A SoftMoE module predicts the next latent state, while a SparseMoE module predicts rewards; a planner then searches in latent space. On the Bi-DexHands and MA-Mujoco suites, M3W improves return and sample efficiency on opposing tasks when compared with recent modular-policy baselines like MACD and HMASD. Visual analyses further show that different experts specialize in intuitive joints and task groups, lending interpretability to the bounded-similarity idea.

**Questions:**

1.	Have the authors conducted comparative experiments using a uniform SoftMoE or uniform SparseMoE setup? If the heterogeneous design is removed, how would the performance and computational cost change?
2.	Table 9 indicates that M3W achieves lower performance on the 6a-Cheetah task. Does this imply that M3W’s performance deteriorates as the number of agents increases?
3.	Could the proposed approach be adapted to partially observable environments? If so, what specific modifications would be necessary?

**Ethical Concerns:**

["NO or VERY MINOR ethics concerns only"]

**Final Justification:**

I appreciate the authors’ thoughtful response in their rebuttal, which effectively clarified my initial concerns and resolved my confusion. Accordingly, I have decided to increase my rating.

**Limitations:**

Yes

**Quality:**

3

**Strengths And Weaknesses:**

Strengths:

1. M3W moves modularity from the policy to an MoE-based world model, allowing tasks that share underlying physics but need different behaviors to reuse dynamics while avoiding the gradient clashes that often sink multitask RL.

2. The paper first introduces the MoE-based world model for multi-task settings, combining a SoftMoE dynamics predictor and a SparseMoE reward predictor. Task-conditioned routing selectively re-uses experts, reducing gradient interference and enhancing interpretability through expert specialization.

3. The paper incorporates a load-balance loss to equalize expert utilization, preventing overload of popular experts and enhancing training robustness, thereby avoiding the expert-imbalance pitfalls observed in early MoE works.
4. Experiments on two challenging benchmarks-Bi-DexHands and MA-Mujoco show that M3W achieves the top return, with improvements significant over model-based and model-free baselines.

5. The paper provides attention distributions of individual experts, cosine-similarity of router outputs revealing task-relatedness, and t-SNE visualizations of latent trajectories—strengthening the  interpretability of M3W.


Weakness:

1. The evaluation primarily focuses on two benchmarks which share similar continuous control settings. It remains unclear whether the concept of bounded similarity highlighted by the authors applies broadly to other types of environments, particularly those with different characteristics.

2. Because the planner relies on MPPI control, M3W is only applicable to tasks with continuous action spaces. Consequently, the method cannot yet be deployed in many popular multi-agent benchmarks such as SMAC or Atari-style games. Moreover, M3W further assumes full observability, restricting applicability in partially observable domains.

3. Even after applying the authors’ early-stopping heuristic, each control step still takes a considerable amount of time on an RTX A6000. Moreover, the paper reports no GPU-time for competing baselines, making direct comparison impossible.

---

> ### Author Rebuttal · Authors · 2025-07-27
>
> `Responses`
> ---
> ---
> > **Q1: Does the concept of bounded similarity generalize to other environments with with different characteristics? (Weakness.1)**
>
> **A1:** We believe that the bounded similarity phenomenon is not limited to continuous-control settings but represents **a general property of multi-task environments.** To further illustrate, we conducted experiments similar (Fig. 1, L27, main text) on other environments.
>
> + **Experimental setting:** We conducted additional experiments on three representative environments discussed in (D.4 *Why Choose Bi-DexHands and MA-Mujoco?*, L541, Appendix) — **SMAC** (discrete actions, partial observability, heterogeneous agents), **MPE** (continuous actions, full observability, homogeneous agents), and **RWARE** (mixed difficulty and partial observability)—covering substantially different action spaces, observability assumptions, and agent heterogeneity.
>
> + **Method:** Task trajectories can be treated as samples of the underlying dynamics. We measured pairwise similarity between task dynamics using Maximum Mean Discrepancy (MMD) [1] between task trajectories (with lower scores indicating higher similarity).
>
> + **Results:** Across all environments, tasks within the same group (e.g., MMM & MMM2) consistently exhibit much lower MMD scores than across-group pairs (e.g., 2s3z & MMM), indicating clear intra-group similarity and inter-group divergence. This confirms that **bounded similarity is consistently observed across diverse environments, validating its generality.**
>
> + For clarity, only the upper triangle of the symmetric MMD matrix is shown:
>
>   SMAC|2s3z|3s5z|MMM|MMM2
>   -|-|-|-|-
>   **2s3z**||**0.15**|0.58|0.69
>   **3s5z**|||0.58|0.69
>   **MMM**||||**0.09**
>   **MMM2**||||
>
>   MPE|adversary|push|reference|spread
>   -|-|-|-|-
>   **adversary**||**<0.01**|0.40|0.38
>   **push**|||0.39|0.38
>   **reference**||||**0.03**
>   **spread**||||
>
>   RWARE|left-easy|left-hard|right-easy|right-hard
>   -|-|-|-|-
>   **left-easy**||**<0.01**|0.13|0.12
>   **left-hard**|||0.14|0.13
>   **right-easy**||||**<0.01**
>   **right-hard**||||
>
> &nbsp;
>
> ---
> > **Q2: Could the proposed approach be extended to partially observable and discrete-action environments, and how? (Weakness.2 & Question.3)**
>
> **A2:** We start from two perspectives: partially observable and discrete actions.
>
> + **Partial Observability:** We acknowledge that the full-observability assumption is indeed a common limitation of existing model-based MARL methods [2,3]. However, M3W can be naturally extended to partial observability by two simple modifications:
>   1. **Training-time masking:** Inspired by Masked AutoEncoder [4] (and its applications in MARL [5]), we randomly mask other agents with a certain probability to simulate communication failures, thereby enhancing robustness.
>   2. **Planning-time communication cache:** The planner caches previously predicted rollouts and reuses them in case of communication failure, replacing missing ground-truth inputs to maintain trajectory continuity.
>
>   We tested the above extensions on *BiDexHands-Over* and confirmed their effectiveness. In *mask* group, other agents are masked with 50% probability, and communication fails with 50% probability during execution. **These results show that M3W can adapt to partial observability with minimal performance degradation.**
>
>   ||full|mask|decentralized|
>   -|-|-|-
>   Dynamics Error ↓|0.025±0.003|0.033±0.005|0.044±0.005
>   Reward Error ↓|0.004±0.001|0.012±0.001|0.013±0.001
>   Return ↑|32.1±0.4|31.3±0.6|31.1±0.3
>
> + **Discrete Action Space:** There are two modifications to extend our method to discrete action spaces:
>   1. Use one-hot actions or bitactions [6] to transform a discrete action into a continuous one.
>   2. Replace the MPPI control with Cross Entropy Method (CEM) or Monte Carlo Tree Search (MCTS).
>
>   During training, discrete actions are converted into one-hot representations to facilitate understanding by the world model. During execution, a CEM-based planner is adopted, and its procedure is reported below. Similar to the above, we evaluated these extensions on the classic discrete tasks SMAC ‘3m’ and RWARE ‘left-easy’, achieving a **100% win rate** and a reward **> 7.75/100 steps**.
>
>   **CEM Procedure:** Assume the action dimension is $n$,
>   + S1: Initialize a discrete action distribution, $p^{(0)}=1/n$
>   + S2: Sample $M$ action sequences, and compute their values according to (Eq.7, L185, main text)
>   + S3: Select the top $m$ elite action sequences $a^∗$
>   + S4: For each action dimension $j$, update the distribution $p^{(k+1)}(a[j])=\frac{1}{m} \sum \mathbb{I}(a[j]=a^*)$
>   + Repeat steps S2–S4 until convergence.
>
> &nbsp;
>
> ---
> > **Q3: Inference time cost of baselines. (Weakness.3)**
>
> **A3:** The table below reports inference time (ms/step) on Bi-DexHands. While policy-centric methods avoid online planning and thus achieve lower inference times at the cost of flexibility, our method remains comparable to, or even faster than, existing planner-based approaches [7,8]. Moreover, our early-stopping heuristic further reduces the inference time to < 10 ms/step, making real-time applications feasible.
>
> ||Ours(default)|Ours(early-stop)|MACD|HMASD|MAT|MAPPO|HATRPO
> |-|-|-|-|-|-|-|-
> Time Cost (ms)|25.8|9.8|1.4|3.4|2.2|1.1|1.1
>
> &nbsp;
>
> ---
> > **Q4: How would the performance and computational cost change if the heterogeneous MoE design is replaced with a uniform one? (Question.1)**
>
> **A4:** The heterogeneous design was chosen for two main reasons:
> 1. **SoftMoE for dynamics modeling:** As noted in [9], SoftMoE is fully differentiable and effectively captures token-wise correlations, making it suitable for modeling complex inter-agent interactions.
> 2. **SparseMoE for reward prediction:** SoftMoE is an n-to-n architecture, whereas reward prediction is an n-to-1 mapping, so we choose SparseMoE.
>
> We additionally conducted experiments replacing the heterogeneous design with uniform SoftMoE and uniform SparseMoE. The results show that uniform SoftMoE increases reward prediction error and time cost, while uniform SparseMoE worsens dynamics modeling. Thus, the heterogeneous design achieves the best trade-off between accuracy and efficiency.
>
> ||Dynamics Error ↓|Reward Error ↓|Time Cost(ms/step) ↓
> -|-|-|-
> Heterogeneous|**0.042±0.007**|**0.147±0.004**|25.8
> Uniform-SoftMoE||0.151±0.004|30.1
> Uniform-SparseMoE|0.063±0.008||**19.2**
>
> &nbsp;
>
> ---
> > **Q5: Does the performance deteriorates as the number of agents increases? (Question.2)**
>
> **A5:** our method achieves second-best performance on 6a-Cheetah. We believe the slight performance drop is not due to scalability limitations, but rather due to limited training samples for 6-agent tasks in the MA-MuJoCo benchmark.
>
> + **Scalability Analysis:** As shown in (E.7 *Scalability with the Number of Agents, L663*, Appendix), we evaluated M3W on *10-agent Swimmer* and *56-agent MAgent*, where it maintained stable performance and outperformed all baselines. This demonstrates that M3W can scale to significantly larger numbers of agents.
> + **Why the Drop on 6a-Cheetah?** In (Section 4.1 *Learning Multi-Agent Dynamics as Sequence Prediction*, L108, main text), we model the multi-agent dynamics as a sequence prediction problem, which makes our approach more sensitive to the number of agents compared to policy-centric methods. Most MA-MuJoCo tasks involve 2–3 agents, and 6-agent tasks are underrepresented, leading to slightly worse sample efficiency.
> + **Future Work:** We plan to further incorporate permutation-invariance module or autoregressive mechanism to better handle tasks with a varying number of agents.
>
> &nbsp;
>
>
> `Summary`
> ---
> ---
>
> Thank you for your valuable review and constructive feedback. We appreciate your recognition of the **motivation**, **novelty**, **strong performance**, and **interpretability** of our work.
>
> We also acknowledge the affirmations from other reviewers R2 (gkLA), R3 (mmFi), and R4 (k53v) on:
>
> + **Motivation and novelty:** (**R2:** *identifies the novel insight of ‘bounded similarity’*; *introduce a MoE within world model is a conceptual shift in MT-MARL*), (**R3:** *well-structured and strongly motivated*), (**R4:** *proposes a new perspective on modularity*).
>
> + **Method design:** (**R2:** *allows better exploitation of inter-task structure;* *modular components selectively share or isolate knowledge*), (**R4:** *MoE world model effectively captures bounded similarity among tasks*).
>
> + **Strong performance:** (**R2:** *outperforms baselines in both return and sample efficiency*), (**R3:** *really strong results*), (**R4:** *comprehensive ablation studies and visualizations*; *superior performance and sample efficiency*).
>
> + **Interpretability:** (**R3:** *interpretability analysis shows the mechanism of sharing experts*).
>
> We hope that our responses have fully addressed your concerns, and we welcome any further suggestions and discussions. **If our responses have resolved your concerns, we would sincerely appreciate your consideration for raising the score.**
>
> &nbsp;
>
>
> `References`
> ---
> ---
>
> [1] A kernel two-sample test, JMLR, 2012.
>
> [2] Aligning Credit for Multi-Agent Cooperation via Model-based Counterfactual Imagination, AAMAS, 2024.
>
> [3] Decentralized Transformers with Centralized Aggregation are Sample-Efficient Multi-Agent World Models, TMLR, 2025.
>
> [4] Masked Autoencoders Are Scalable Vision Learners, CVPR, 2022.
>
> [5] MA2E: Addressing Partial Observability in Multi-Agent Reinforcement Learning with Masked Auto-Encoder, ICLR, 2025.
>
> [6] INS: Interaction-aware Synthesis to Enhance Offline Multi-agent Reinforcement Learning, ICLR, 2025.
>
> [7] TD-MPC2: Scalable, Robust World Models for Continuous Control, ICLR, 2024.
>
> [8] Sparse Imagination for Efficient Visual World Model Planning, arXiv, 2025.
>
> [9] From Sparse to Soft Mixtures of Experts, ICLR, 2024.

---

> > ### Comment · Reviewer_DXnW · 2025-08-07
> >
> > I appreciate the authors’ thoughtful response in their rebuttal, which effectively clarified my initial concerns and resolved my confusion.  Accordingly, I have decided to increase my rating.

---

### Note · Authors · 2025-08-11

We sincerely thank all reviewers and the AC for their time, constructive feedback, and active engagement during the rebuttal and discussion phases. We are especially honored that three reviewers (R1/DXnW, R2/gkLA, R4/k53v) explicitly stated that their concerns had been resolved and that they would raise their scores. We also appreciate the constructive suggestions from R3/mmFi.

Our proposed method, M3W, introduces a Mixture-of-Experts (MoE) based world model, motivated by novel concept of bounded similarity in multi-task dynamics. By shifting modularity from the policy to the world model, M3W  achieves superior performance, sample efficiency, and interpretability on two challenging MT-MARL benchmarks.

We are encouraged by the reviewers’ consistent recognition of our contributions:

+ **Novelty & Motivation:** Introducing MoE within the world model as a conceptual shift, and identifying bounded similarity as an important property.

+ **Method Design:** Task-conditioned expert routing and a heterogeneous MoE structure.

+ **Strong Results:** Significant improvements in return and sample efficiency, supported by comprehensive ablations and visual analyses.

+ **Interpretability:** Clear expert specialization aligned with intuitive task groups.

During the rebuttal and discussion, we made extensive efforts to address all raised concerns:

+ **Generality of bounded similarity:** Conducted additional experiments on SMAC, MPE, and RWARE, confirming its presence across diverse environments.

+ **Extension to partial observability and discrete actions:** Proposed and validated simple yet effective modifications (training-time masking, planning-time communication cache, discrete-action planners).

+ **Computation and scalability:** Reported detailed inference-time comparisons, analyzed scalability to large-agent settings, and justified the heterogeneous MoE design via ablations.

+ **Clarifications:** Updated loss formulations for transparency and clarified implementation details.

We deeply value the constructive rebuttal and discussion process. We believe that these additional experiments, clarifications, and extensions have further strengthened the paper and demonstrated the broad applicability and robustness of M3W. We are committed to incorporating all improvements into the camera-ready version to ensure a complete and rigorous presentation.

---

### Decision · Program_Chairs · 2025-09-17

**Decision:**

Accept (poster)

**Comment:**

This paper introduces M3W, a mixture-of-experts world model for multi-task multi-agent reinforcement learning. The core conceptual contribution is shifting modularity from policies to the world model, motivated by the idea of bounded similarity across task dynamics. The method combines a SoftMoE dynamics predictor and a SparseMoE reward predictor, with task-conditioned routing to enable selective sharing of knowledge across related tasks. Planning is performed using imagined rollouts from the world model rather than relying on an explicit policy.

The strengths are clear. Reviewers consistently recognized the novelty of applying MoE modularity within the world model, which represents a meaningful conceptual shift in MT-MARL. The empirical evaluation on Bi-DexHands and MA-MuJoCo is strong, with consistent improvements in return and sample efficiency compared to policy-centric baselines. The interpretability analyses (expert specialization, similarity matrices, visualizations) provide additional insight into the bounded similarity phenomenon. The rebuttal further strengthened the work by supplying ablations (heterogeneous vs. uniform MoE, number of experts, actor-only vs. planner-only), extensions to partial observability and discrete action spaces, and additional results on scalability and computational cost. These clarifications satisfied reviewers, three of whom raised their scores after rebuttal.

The main limitations are scope and practicality. Evaluation is limited to continuous-control multi-agent benchmarks, leaving open how well the approach translates to discrete or partially observable domains. While the authors provided preliminary extensions, further validation in diverse environments would strengthen claims of generality. The method also carries higher computational cost compared to policy-based baselines, though early-stopping mitigates this to some extent. Finally, the related work discussion, particularly on prior MoE applications in MTRL, was initially thin but will be expanded in the camera-ready version.

Overall, this is a technically solid and well-motivated contribution. The paper offers a novel perspective on modularity, demonstrates strong empirical results, and provides thoughtful analyses and extensions. The limitations are acknowledged and do not outweigh the contributions.